# Education, financial aid and awareness can reduce smallholder farmers' vulnerability to drought under climate change

Marthe L.K. Wens[1], Anne F. van Loon[1], Ted I.E. Veldkamp[2], Jeroen C.J.H. Aerts[1]

[1]Institute for Environmental Studies, Vrije Universiteit Amsterdam, the Netherlands
[2]Urban Technology, Amsterdam University of Applied Sciences, The Netherlands

*Correspondence to*: Marthe Wens (marthe.wens@vu.nl)

**Abstract.** Analyses of future agricultural drought impacts require a multidisciplinary approach in which both human and environmental dynamics are studied. In this study, we used the socio-hydrologic, agent-based drought risk adaptation model ADOPT. This model simulates the decisions of smallholder farmers regarding on-farm drought adaptation measures, and the resulting dynamics in household vulnerability and drought impact over time. We applied ADOPT to assess the effect of four top-down disaster risk reduction interventions on smallholder farmers' drought risk in the Kenyan drylands: The robustness of additional extension services, ex-ante rather than ex-post cash transfers, improved early warnings and lowered credit rates was evaluated under different climate change scenarios.

Model results suggest that extension services increase the adoption of low-cost, newer drought adaptation measures while credit schemes are useful for measures with a high investment cost, and ex-ante cash transfers allow the least wealthy households to adopt low-cost well-known measures. Improved early warning systems show more effective in climate scenarios with less frequent droughts. Combining all four interventions displays a mutually-reinforcing effect with a sharp increase in the adoption of on-farm drought adaptation measures resulting in reduced food insecurity, decreased poverty levels and drastically lower need for emergency aid, even under hotter and drier climate conditions. These nonlinear synergies indicate that a holistic perspective is needed to support smallholder resilience in the Kenyan drylands.

**Key words:** Agent-based modelling, drought disasters, risk reduction, adaptation measures, adaptive behaviour, smallholder farmers, drought adaptation, AquacropOS, ADOPT, risk assessment; Kenya, dryland agriculture

## 1 Introduction

Droughts, defined as below-normal meteorological or hydrological conditions, are a pressing threat to the food production in the drylands of Sub-Saharan Africa (Brown et al., 2011; Cervigni & Morris, 2016; UNDP et al., 2009). Over the last decades, increasing temperatures and erratic or inadequate rainfall have already intensified drought disasters (Khisa, 2017). Climate change, population growth and socio-economic development will lead to additional pressures on water resources (Erenstein, Kassie, & Mwangi, 2011; Kitonyo et al., 2013). In Kenya, three quarters of the population depend on smallholder rain-fed agricultural production and nearly half of the population is annually exposed to recurring drought disasters causing income insecurity, malnutrition and health issues (Alessandro et al., 2015; Khisa, 2018; Mutunga et al., 2017; Rudari et al., 2019; UNDP, 2012). Reducing drought risk is imperative to enhance the resilience of the agriculture sector, to protect the livelihoods of the rural population, and to avoid food insecurity and famine in Kenya's drylands (Khisa, 2017; Shikuku et al., 2017).

Drought risk models are important tools to inform policy makers about the effectiveness of adaptation policies and enable the design of customized drought adaptation strategies under different future climate scenarios (Carrao et al., 2016; Stefano et al., 2015). Traditionally, such models express disaster risk as the product of hazard, exposure and vulnerability, and are based on historical risk data. Recent disaster risk models have dealt with climate change adaptation in a two-stage framework; first describing a few scenarios regarding adaptation choices of representative households, then estimating the impacts of adaptation on (future-) welfare while assuming climate change scenarios (di Falco, 2014). However, most existing research does not account for more complex dynamics in adaptation and vulnerability (Conway et al., 2019), for the heterogeneity in human adaptive behaviour (Aerts et al. 2018) or for the feedback between risk dynamics and adaptive behaviour dynamics (Di Baldassarre et al., 2017). Though, these are the aspects that determine, for a large part, the actual risk (Eiser et al., 2012).

It appears that farmers often act boundedly rational towards drought adaptation rather than economically rational: their economic rationality is bounded in terms of cognitive capability, information available, perceptions, heuristics and biases (Schrieks et al., 2021; Wens et al., 2021). To account for such individual adaptive behaviour in drought risk assessments, an agent-based modelling technique can be applied (Berger & Troost, 2014; Blair & Buytaert, 2016; Filatova et al., 2013; Kelly et al., 2013; Matthews et al., 2007; Smajgl et al., 2011; Smajgl & Barreteau, 2017). Agent-based models allow explicit simulation of the bottom-up individual human adaptation decisions and capture the macro-scale consequences that emerge from the interactions between individual agents and their environments. Combining risk models with an agent-based approach is thus a promising way to analyse drought risk, and the evolution of it through time, in a more realistic way (Wens et al., 2019).

Here we present how an agent-based drought risk adaptation model, ADOPT (designed in Wens et al 2020), can increase our understanding of the effect of drought policies on community-scale drought risk for smallholder farmers in Kenya's drylands.. The design of ADOPT as an agent-based drought risk adaptation model is described in Wens et al., 2020. Moreover, Wens et al. (2021) detail the empirical data on past adaptive behaviour (used to calibrate the model), as well as empirical data on adaptation intentions that can be used to compare with the model outputs.

In this study, we apply the ADOPT model, to test the variation in household drought risk under different drought management policies: (i) a reactive government only providing emergency aid, (ii) a pro-active government, which provides sufficient drought early warnings and ex-ante cash transfer in the face of droughts , and (ii) a prospective government that, in addition to early warnings and ex-ante transfers, subsidises adaptation credit schemes and provides regular drought adaptation extension services to farmers. In addition, ADOPT is used to evaluate the robustness of these policies under different climate change scenarios. We acknowledge that ADOPT should be subject to additional validation steps in order to more accurately and precisely predict future drought risk. Yet, in this study we elaborate the potential of this proof-of-concept model by showcasing the trends in drought risk under risk reduction interventions and climate change for a case study in semi-arid Kenya.

## 2 Case study description

The ADOPT model has been applied to the context of smallholder maize production in the dryland communities in the areas Kitui, Makueni and Machakos in south-eastern Kenya (fig. 1). This semi-arid to sub-humid region is drought-prone, being hit by drought disasters in 1983/84, 1991/92, 1995/96, 1998/2000, 2004/2005, and 2008-11, 2014-2018 (data from Em-Dat and DesInventar). The majority of the population in this dry transitional farming zone is directly or indirectly employed through agriculture. However. technology adoption and production level remain rather low, making the region very vulnerable to droughts and climate change (Khisa & Oteng, 2014; Mutunga et al., 2017).

In Kenya, 75% of the country's maize is produced by smallholder farms. Maize is grown in the two rainy seasons, with the aim to meet household food needs (subsistence farming) (Erenstein, Kassie, & Mwangi, 2011; Erenstein, Kassie, Langyintuo, et al., 2011; Speranza et al., 2008). While during the long rainy season (March-April-May) multiple crops are planted, the short rainy season (October-November-December) is considered the main growing season for maize in the region (Rao et al., 2011).

Reported smallholder maize yields often do not exceed 0.7 ton/ha. However, with optimal soil water management, maize yields can easily be around 1.3 ton/ha in the semi-arid medium potential maize growing zone in south-eastern Kenya (Omoyo et al., 2015). Few farmers use pesticides or improved seeds or other adaptation strategies (Tongruksawattana & Wainaina, 2019) . In Kitui, Makueni and Machakos, the most preferred seed-variety is the high yielding but less drought resistant Kikamba/Kinyaya variety (120 growing days) with a potential yield of only 1.1 tons per hectare (Speranza, 2010; Recha et al., 2012). Trend analysis (1994-2008) shows that yields are declining due to the increasing pace of recurring droughts (Nyandiko, 2014).

Over 97% of the smallholder farmers in this area grow maize, mainly for own consumption or local markets (Brooks et al., 2009; Kariuki, 2016; Nyariki & Wiggins, 1997). It is the main staple food, providing more than a third of the caloric intake, and is also the primary ingredient used in animal feeds in Kenya (Adamtey et al., 2016; FAO, 2008). .. Only about 20% of the farmers are able to sell their excess crops, while 66% have to buy maize to complement their own production (Muyanga, 2004).

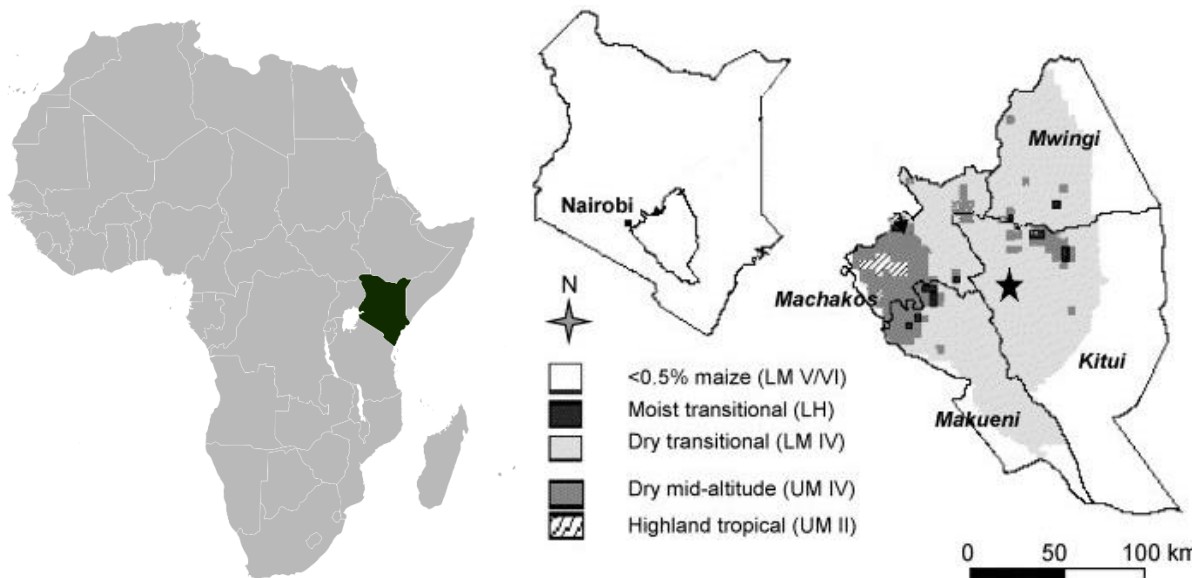

Figure 1: Study area: dry transitional maize agro-ecological zone (right) located in South-Eastern Kenya (centre) in the Horn of Africa (left). Area where the survey data (Wens 2021) is collected is indicated with a star on the right map. Map adjusted from Barron and Okwach (Barron & Okwach, 2005)

**3 Model and scenario description**

ADOPT (fig. 2, Wens et al 2020, ODD+D (Overview, Design concept, Details + Decision) protocol in Appendix A) is an agent-based model that links a crop production module to a behavioural module evaluating the two-way feedback between drought impacts and drought adaptation decisions. ADOPT was parameterized with information from expert interviews, a farm household survey with 260 households including a semi-structured questionnaire executed in the Kitui Region, Kenya (Wens et al. 2021). Moreover, a discrete choice experiment (a quantitative method to elicit preferences from participants without directly asking them to state their preferred options) was executed to get information on changes in adaptation intentions under future top-down DRR interventions (Wens et al. 2021). This empirical dataset feeds the decision rules in ADOPT describing farm households' adaptive behaviour in the face of changing environmental conditions (drought events), social networks(actions of neighbouring farmers), and top-down interventions (drought management policies). In ADOPT, crop production is modelled using AquacropOS (Foster & Brozović, 2018), simulating crop growth on a daily basis and producing crop yield values at harvest time twice per year. Calibrated for the Kenyan dryland conditions (Ngetich et al., 2012; Wamari et al., 2007), AquacropOS considers the current water management of the farm (i.e., the applied drought adaptation measures) and yields vary with weather conditions. The adaptive behaviour of the farm households (agents) is modelled based on the Protection Motivation theory (PMT, Rogers 1975). This theory was derived as promising in an earlier study (Wens et al, 2020) and includes multiple relevant factors that drive the observed behaviour of farm households (Wens et al 2021). In this application of ADOPT, the model was run over 30 historical years as baseline followed by 30 years of future scenarios (combinations of policy and climate changes; the start of these changes is indicated as "year 0"). Through a sensitivity analysis, both the average effect of individual adaptation decisions and its endogenous model variability are analysed

(similar to Wens et al 2020). We used 12 different initialisations per scenario to include variations in model
initialisation, the stochasticity that determines the individual adaptation decisions, and the relative weights of
factors influencing behaviour (See 3.1).

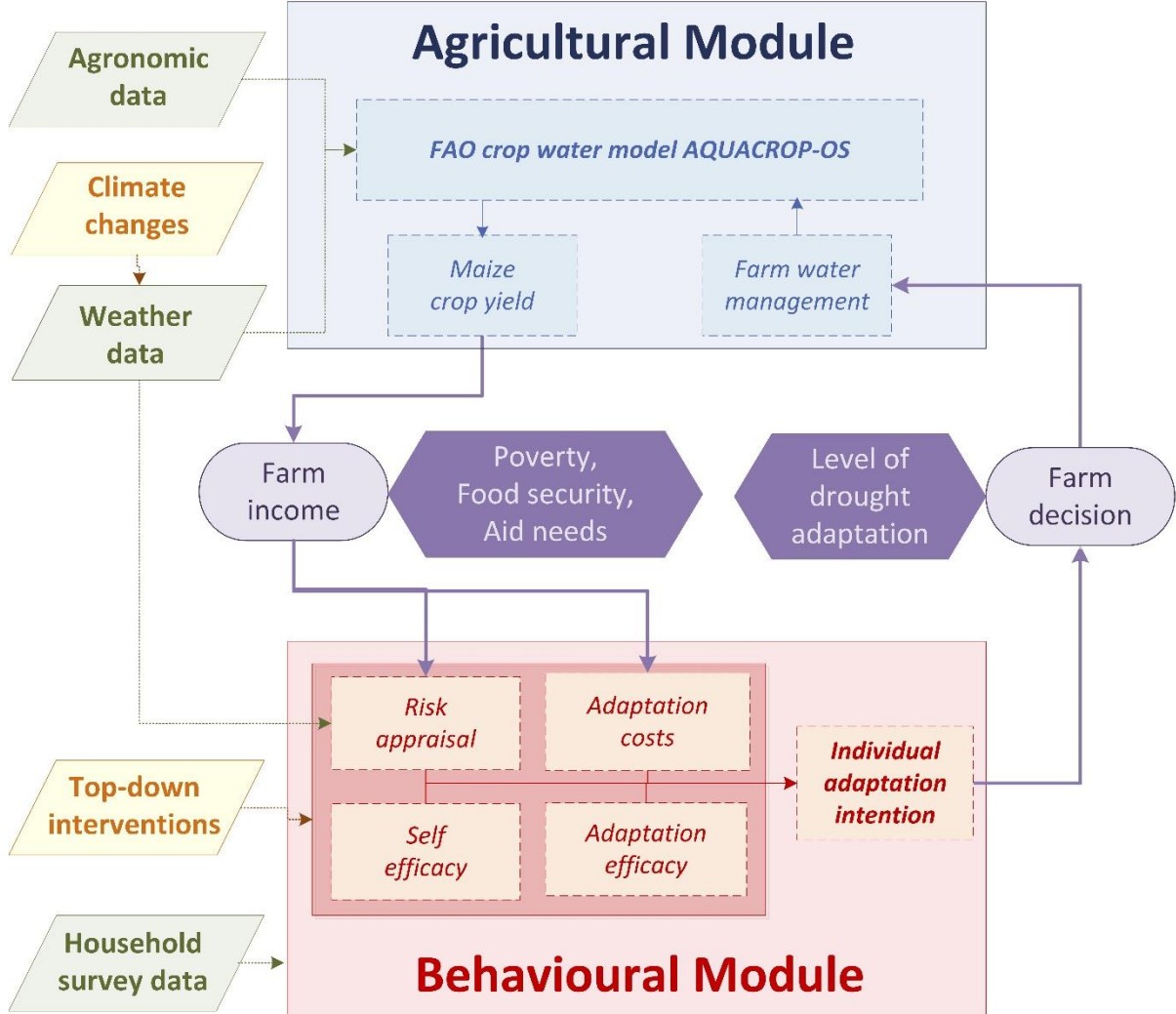

**Fig. 2: ADOPT model overview, adjusted from** Wens et al., 2020**. Description of the model (Overview, , Design concepts**
**& Details) in Appendix A.**

### 3.1 Individual adaptive behaviour in ADOPT

Various soil water management practices, further called drought adaptation measures, can be adopted by
smallholder farmers in ADOPT. There are shallow wells to provide irrigation water, the option to connect these
to drip irrigation infrastructure, and Fanya Juu terraces as on-farm water harvesting techniques. Moreover, a soil
protection measure reducing the evaporative stress, mulching, is included. These measures are beneficial in most
– if not all – of the years and have a particularly good effect on maize yields in drought years. Nonetheless, current
adoption rates of these measures are quite varied and often remain rather low (Gicheru, 1990; Kiboi et al., 2017;
Kulecho & Weatherhead, 2006; Mo et al., 2016; S. Ngigi, 2019; S. N. Ngigi et al., 2000; Rutten, 2004; Zone,
136 2016).

ADOPT applies the Protection Motivation Theory, a psychological theory often used to model farmer's bounded rational adaptation behaviour (Schrieks et al 2021). It describes how individuals adapt to shocks such as droughts and are motivated to react in a self-protective way towards a perceived threat (Grothmann & Patt, 2005; Maddux & Rogers, 1983). Four main factors determining farmers' adaptation intention under risk are modelled: (1) risk perception is modelled through the number of experienced droughts and number of adopted measures, household vulnerability, and experienced impact severity. Moreover, trust in early warnings is added, which can influence the risk appraisal if a warning is sent out. Coping appraisal is modelled through a (2) farmers' self-efficacy (household size / labour power, belief in God, vulnerability), (3) adaptation efficacy (perceived efficiency, cost and benefits, seasons in water scarcity, choices of neighbours, number of measures), and (4) adaptation costs (farm income, off-farm income, adaptation spending, access to credit). These four PMT factors receive a value between 0 and 1 and define a farmer's intention to adopt. Which smallholder farmers adopt which measures in which years is then stochastically determined based on this adaptation intention. More information regarding the decision making can be found in Appendix A.

## 3.2    Drought risk indicators in ADOPT

In ADOPT, annual maize yield influences the income and thus assets of the (largely) subsistence farm households. This influence is indirect, because the farm households are assumed to be both producers and consumers, securing their own food needs. The influence is also a direct one, because these farm households sell their excess maize on the market at a price sensitive to demand and availability. Farm households who cannot satisfy their food needs by their own production, turn to this same market. They buy the needed maize – if they can afford it and if there is still maize available on the market. If they do not have the financial capacity or if there is a market shortage, they are deemed to be food insecure. Their food shortage (the kilogram maize short to meet household food demand) is multiplied by the market price to estimate their food aid needs. Adding the farm income of the household with their income from potential other sources of income, it is estimated whether they fall below the poverty line of 1.9 USD per day. As climate and weather variability causes maize yields to fluctuate over time, so do the prevalence of poverty, the share of households in food insecurity and the total food aid needs. These factors can be seen as proxies for drought risk and were evaluated over time.

## 3.3    Climate change scenarios

Multiple climate change scenarios – all accounting for increased atmospheric carbon dioxide levels - were tested: a rising temperature of 10%, a drying trend of 15%, a wetting trend of 15%, and various combinations of these. The warming and drying trends were based on a continuation of the trends observed in the last 30 years of daily NCEP temperature (Kalnay et al., 1996) and CHIRPS precipitation (Funk et al., 2015) data (authors' calculations; similar trends found in (Gebrechorkos et al., 2020)). The wetting trend was inspired by the projections from most climate change models which predict an increase in precipitation in the long rain season – a phenomenon known as the 'East African Climate Paradox'(Gebrechorkos et al., 2019; Lyon & Vigaud, 2017; Niang et al., 2015). The no change scenario was a repetition of the baseline period, without changing precipitation or temperature hence

only elevated carbon dioxide levels. Reference evaporation was calculated for each scenario using the Penman-
Monteith model and thus influenced by temperature changes (Allen, 2005; Droogers & Allen, 2002).
**Table 1: Average (daily temperature, annual precipitation) weather conditions (1980-2010) in ADOPT**

| | min temperature | max temperature | precipitation | reference evaporation |
|---|---|---|---|---|
| **No change** | 16.3 (+- 0.8) *C | 26.9 (+- 0.9) *C | 888 (+-319) mm | 1547 (+-298) mm |
| **Wet** | 16.3 (+- 0.8) *C | 26.9 (+- 0.9) *C | 1021 (+-367) mm | 1547 (+-298) mm |
| **Hot** | 17.9 (+- 0.9) *C | 29.6 (+- 0.9) *C | 888 (+-319) mm | 1659 (+-320) mm |
| **Dry** | 16.3 (+- 0.8) *C | 26.9 (+- 0.9) *C | 755 (+-271) mm | 1547 (+-298) mm |

These trends were added to time series of 30 years of observed data. While such approach does not account for
an increased variability, it allows to account for the temporal coherence in the data and the interrelationships
among different weather variables (weather generators – another option to downscale projected climate - have
still some progress to make in order to accurately account for extreme events (Ailliot et al., 2015; Mehan et al.,
2017)). This resulted of 30 years of synthetic 'future' data, for each of the six - wet, hot-wet, hot, dry, hot-dry and
no change - scenarios . While they not have a known probability of occurring, they enable testing the robustness
of the on-farm adaptations and top-down drought disaster risk reduction strategies under changing average hydro-
meteorological conditions.

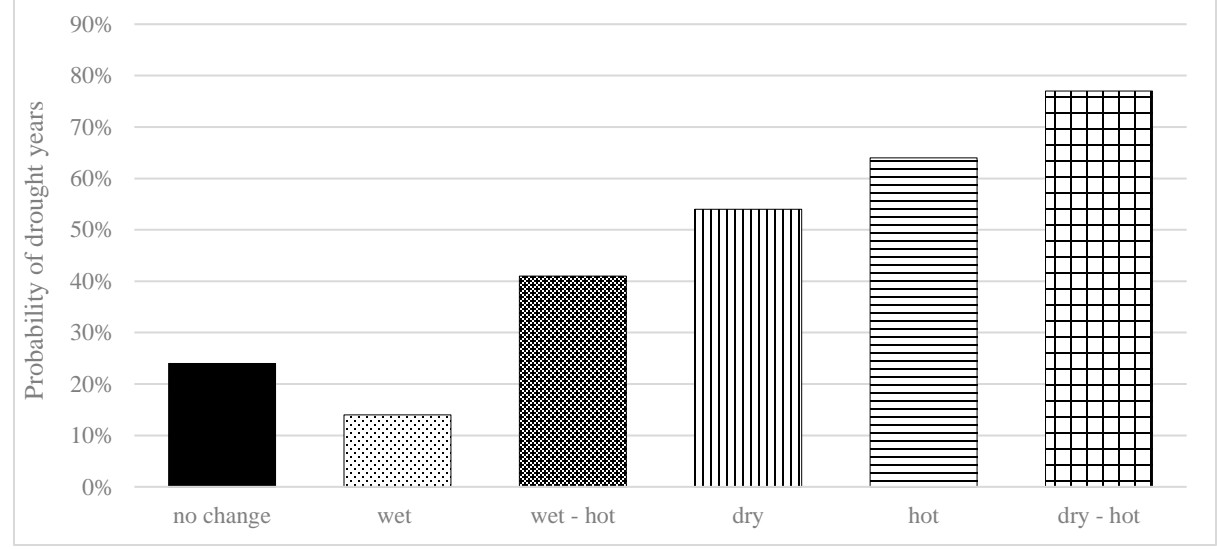

**Fig. 3: Probability of having a year with three or more consecutive months under a SPEI < -1, for the climate change**
**scenarios.**
Droughts, here defined as at least three months with standardized precipitation index (SPEI) values below – 1 ,
have a different rate of occurrence under these different future climate scenarios (Fig. 3). SPEI is calculated
through standardizing a fitted GEV distribution over the historical monthly time series and superimposing this
onto the climate scenario time series. Under the no change scenario, 25% of the thirty simulated years  fall below
this  threshold. Under the wet scenario, fewer droughts occur (15% of the years), but under the dry scenario, the
number of droughts years more than doubles (54% of the years). Temperature is dominant over precipitation is
determining drought conditions, as under the hot-wet scenario, 41% drought years are recorded, and  under hot-
dry conditions, 78% of the years can be considered drought years.

## 3.4 Drought risk reduction intervention scenarios

Kenya Vision 2030 for the ASAL promotes drought management through extension services and aims to increase access to financial services such as affordable credit schemes (Government of Kenya, 2012; Kenya, 2016). Besides, building on the Ending Drought Emergencies plan, the National Drought Management Authority prioritizes the customization, improvement and dissemination of drought early warning systems. It aims to establish trigger levels for ex-ante cash transfer so as to upscale drought risk financing (Government of the Republic of Kenya, 2013; National Drought Management Authority, 2015; Republic of Kenya, 2017). Improved extension services tailored to the changing needs of farm households (Muyanga & Jayne, 2006), a better early warning system with longer lead times (Deltares, 2012; van Eeuwijk, n.d.), ex-ante cash transfers to the most vulnerable when a drought is expected (Guimarães Nobre et al., 2019) and access to credit-markets (Berger et al., 2017; Fan et al., 2013) are all assumed to increase farmers' intention to adopt new measures.

As shown in Wens et al (2021), extension services are best offered to younger, less rich and less educated people, or to those who already adopted the most common measures. Similarly, early warning systems are appreciated more by less educated, less rich farmers, or those not part of farmer knowledge exchange groups. The ex-ante cash transfer instigates those who spend already a lot of money on adaptation, to adopt more expensive measures the most. Access to credit is preferred by less rich farmers, who have a larger land size, are members of a farm group, went to extension trainings, have easy access to information and/or are highly educated (Wens et al. 2021). In this application of ADOPT, the effect of these four interventions - extension services, early warning systems, ex-ante cash transfer and credit schemes - were tested individually. Additionally, three scenarios, combining different types of interventions, were evaluated, all initiated in year "0" in the model run.

1. Reactive policy intervention "supporting drought recovery": Emergency aid is given to farmers who lost their livelihoods after drought disasters; this food aid is distributed to farmers who are on the verge of poverty to avoid famine.

2. Pro-active policy intervention plan "preparing for drought disasters": Early warnings are sent out each season if a drought is expected. This is assumed to raise all farmers' risk appraisal with 20%. Ex-ante cash transfers are given to all smallholder farmers (those without income off-farm and without commercialisation) to strengthen resilience in the face of a drought. This is done when severe and extreme droughts (SPEI <-1, and <-1.5) are expected that could lead to crop yield lower than respectively 500kg/ha and 300kg/ha. Money equivalent to the food insecurity following these yields is paid out to farmers with low external income sources. Moreover, like in the reactive government scenario, emergency aid is given to farmers who need it.

3. Prospective policy intervention plan (UNDRR 2021) "mitigating (future) drought disasters": Credit rates are lowered so that it is affordable to people to take a loan for adaptation measures, at an interest rate of 2% and a pay-back period of five years. Besides, frequent trainings are given in communities with poor practices to improve their capacity related to drought adaptation practices for agriculture. Moreover, like in the proactive government scenario, an improved early warnings system is set up and ex-ante cash transfer is given. Lastly, emergency aid is given to farmers who need it.

### 4.    Results

### 4.1    Maize yield under different adaptation measures and future climate scenarios

The annual average maize yields under the different climate scenarios, for the four on-farm drought adaptation measures implemented in ADOPT - mulch, Fanya Juu bunds, shallow well and drip irrigation -, were calculated using AquacropOS (Fig. 4). Under wetter future climate conditions, maize yields are expected to increase under all management scenarios, with mulch having a particular positive effect on the soil moisture conditions throughout the full growing season. Hotter climate conditions reduce yields slightly:  the assumptions in this model on the frequency and amount of manual irrigation or drip irrigation water are not sufficient to diminish this effect, even under wetter conditions. Paired with drier conditions, this hotter future has dramatically negative effects on yields, showing on average 28% lower yields compared to the no climate change scenario over all management scenarios.

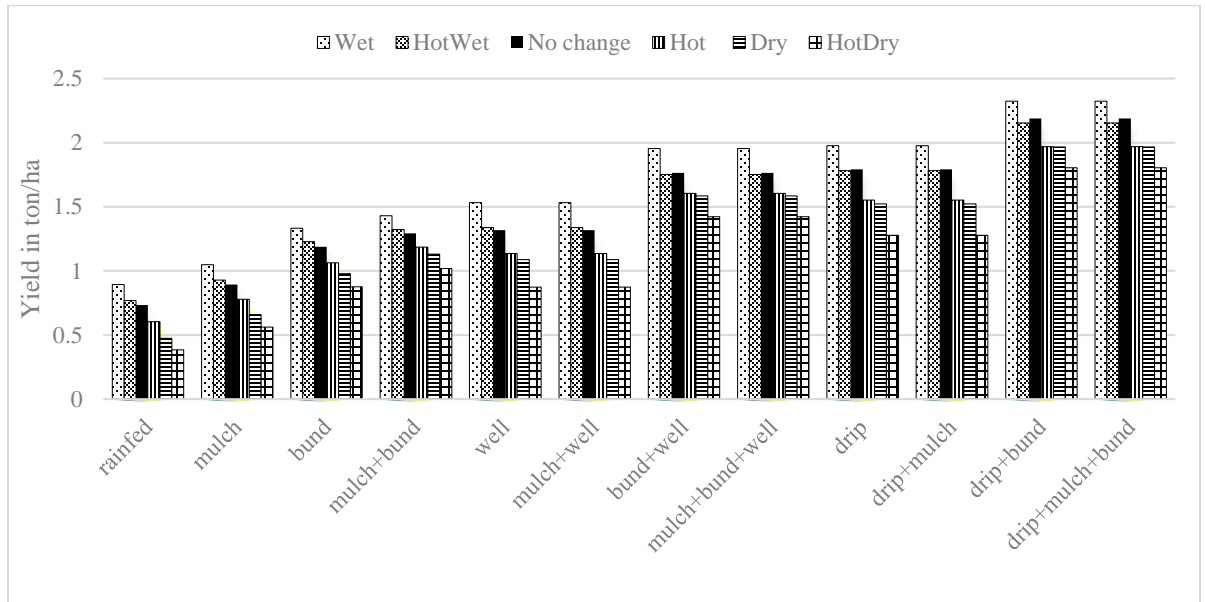

**Fig. 4: Average maize yield under different drought adaptation measures and different future climate scenarios.**

### 4.2    The adoption of adaptation measures over time

In ADOPT, all evaluated top-down interventions increased the adoption rate of the evaluated adaptation measures compared to the reactive "no intervention" scenario (Fig.5): reduced credit rates, improved early warning systems, tailored extension services, and ex-ante cash transfers, as well as the proactive and prospective scenarios  lead to increases in adoption as compared to the reactive scenario (colours in Fig. 5).

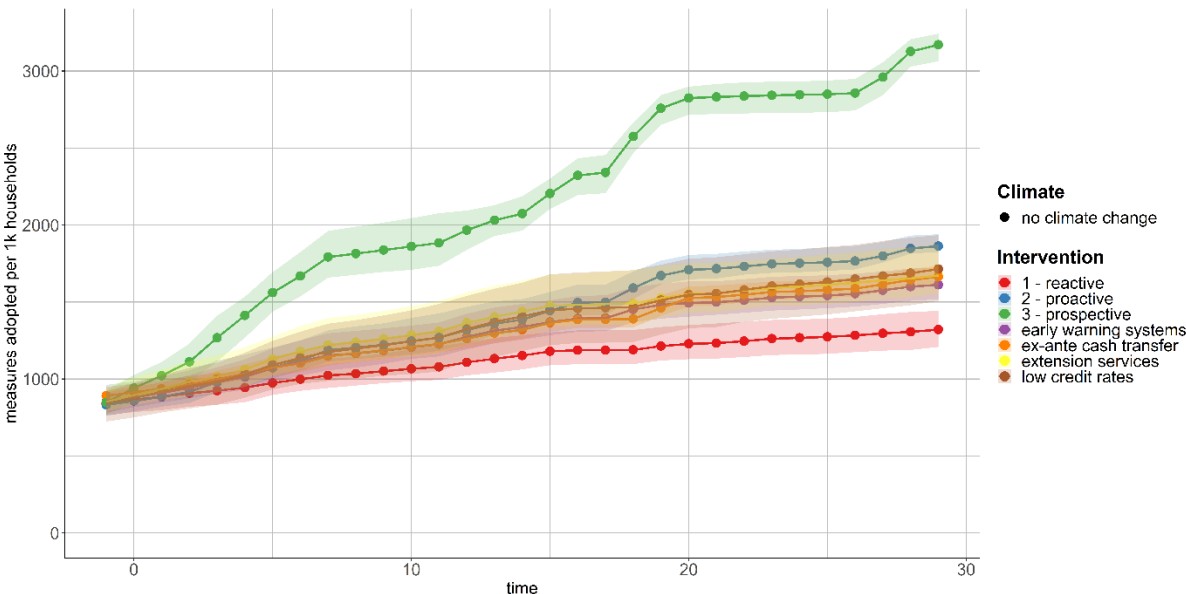

249

**Fig. 5: Total amount of measures adopted per 1000 initialized households under no climate change, averaged over all runs. The shaded area indicates the variation - uncertainty introduced by different model initialisations and by different relative importance of the PMT factors on the decisions of households (sensitivity analysis). Year 0 initiates policy drought risk reduction interventions (indicated with different line colours).**

Looking into detail to the effect of possible policy interventions (Fig. 5, table B2 in Appendix B), affordable credit schemes had the highest effect on the adoption rate of drought adaptation measures. Furthermore, ex-ante cash transfers (which cannot be seen as large sums of investment money but as a mere means to keep families food secure) were more effective to increase adoption of the more affordable measures. Indeed, richer families mostly had already adopted these measures before policy interventions were in place. Extended extension service training increased the adoption of less popular measures and decreased the adoption of the popular but not as cost-effective Fanya Juu terraces. Early Warning Systems had more effect in the wetter climate conditions. The dry-hot scenario has so many drought episodes that risk perception is automatically high while the alert lowers when droughts become scarcer in the less dry scenarios.

Overall, although the processes through which the interventions support households to adapt differ significantly, the differences in eventual adoption rate under the different interventions were small (they overlap in uncertainty interval). Also, the effect of climate change on the adoption rate (Figure B1, Table B2 in Appendix B) was rather small when evaluating the reactive (no intervention) scenario. However, with interventions, the climate change scenarios differed more.

When examining the effect of the three intervention scenarios (Figure B2 in Appendix B; table B2 in Appendix B), it is clear that implementing multiple policies at once resulted in a stronger increase in adoption: a proactive and prospective intervention plan increased the adoption of different adaptation measures with respectively 40% and 140% more than under the "reactive, no climate change" scenario where no intervention takes place. Both a proactive and prospective approach increased the adoption of cheaper adaptation measures to close to 100% of the farm households. For the more expensive measures, the proactive scenario showed to be less effective while the prospective scenario reached quite high adoption rates in the more extreme climate scenarios.

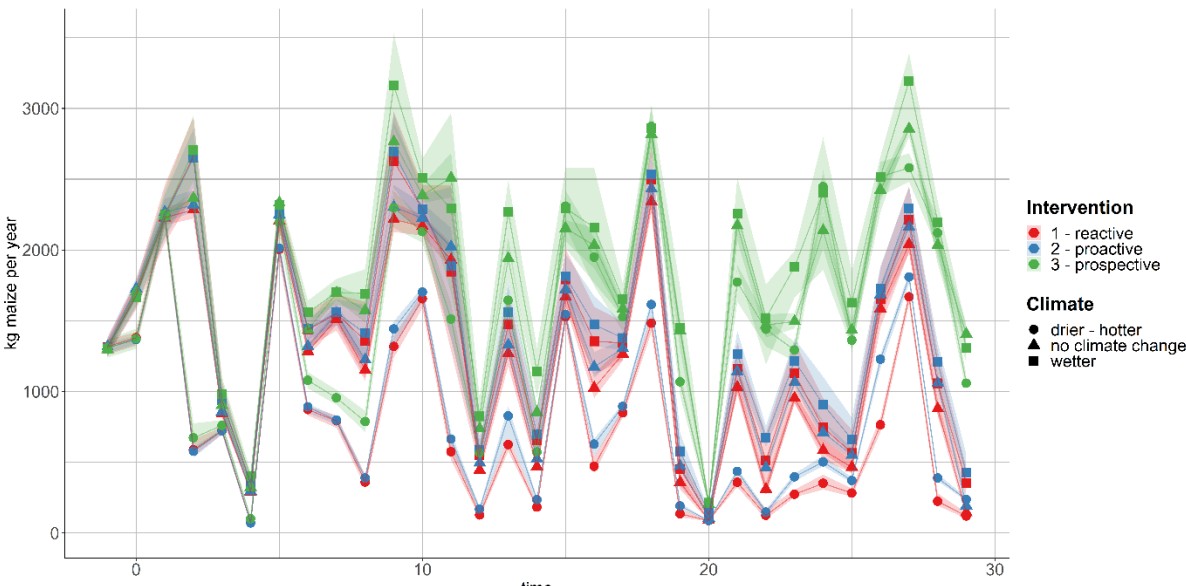

**Fig. 6: Household maize harvest (kg/year, sum of two growing seasons) over 30 'scenario years' under different climate**
**change and policy intervention scenarios. The shaded area indicates the variation - uncertainty introduced by different**
**model initialisations and by different relative importance of the PMT factors on the decisions of households (sensitivity**
**analysis)**
The adoption of adaptation measures by households influenced their maize yield and thus affected the average
and median maize harvest under the different future climates and drought risk reduction interventions (Fig. 6).
This becomes clear comparing the first thirty baseline years with the following thirty scenario years: When no
policy interventions were in place, average maize yields increased with almost 30% under a wet-hot future and
decreased over 25% under a dry-hot climate. Under a prospective government supporting the adoption of
adaptation measures, average maize yields increased up to 100% under a wet-hot future and increased with over
60% under dry-hot future conditions. Clearly, an increased uptake of measures under this intervention scenario
did offset a potentially harmful drying climate trend.
**4.3 Drought risk dynamics under policy and climate change**
Assuming off-farm income to fluctuate randomly but not steadily increasing or decreasing, the changing harvests
over time directly affected the poverty rate and the share of households in food insecurity (Fig. 7). Both trends in
yield caused by droughts or by the adoption of new adaptation measures, could drive farm household in or out of
poverty. Running ADOPT with a reactive and no climate change scenario, a slight increase of 5 percentage points
(pp) in poverty levels was visible. Poverty levels increased up to 15pp compared to the baseline situation, when
a dryer and/or hotter climate scenario was run. A proactive intervention plan reduced poverty by 11pp under no
climate change. In the dry-hot climate scenario this combination of improved early warning systems and ex-ante
cash transfers lead to reductions of 20-30pp compared to the baseline years. However, the prospective government
scenario showed the most prominent results, projecting reductions of 45pp under no climate change and around
60pp under dryer and hotter climate conditions.

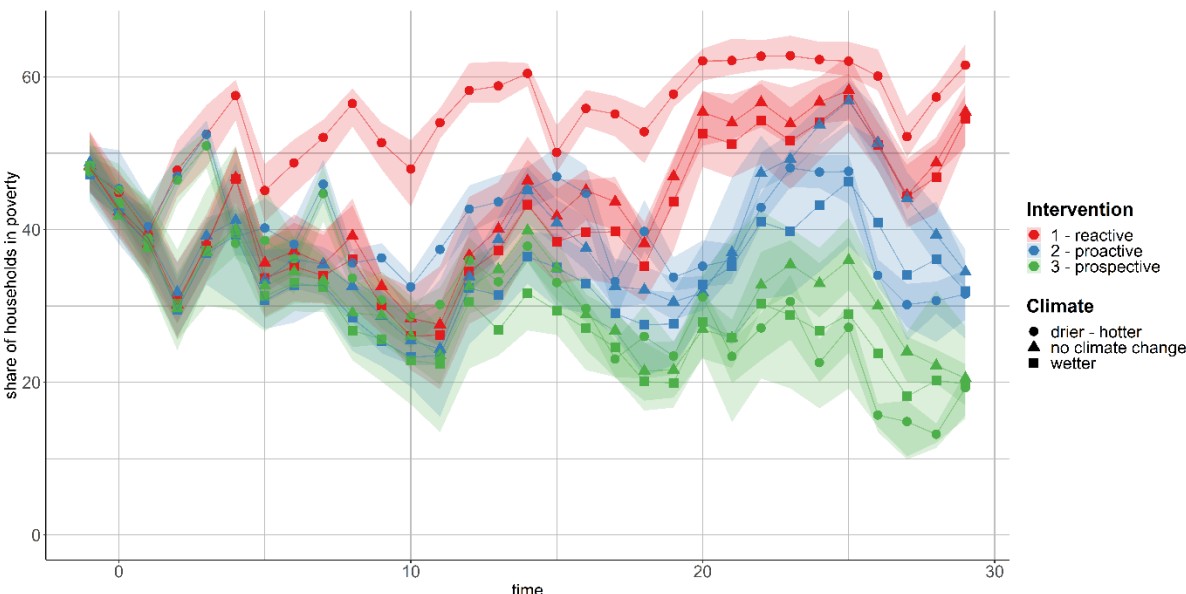


**Fig. 7: Share of households in poverty (earning under the 2USD/day income line, under different climate and policy intervention scenarios). The shaded area indicates the variation - uncertainty introduced by different model initialisations and by different relative importance of the PMT factors on the decisions of households (sensitivity analysis).**

Food insecurity is partly caused by a lack of income or assets, but also by the farm market mechanism. Droughts, climate change and adaptation levels influence the availability of maize on this market. Farm households which do not produce enough to be self-sufficient, buy maize on the market if they have the money and if there is maize locally available. Households are assumed to be in food shortage if they have to rely on food aid to fulfil their caloric needs. On average in the 'no climate change' and 'no policy interventions' scenarios, food security rates were predicted to remain stable compared to the baseline period (fig. 8). However, policy interventions and climate change can alter this balance.

Improving extension services or providing ex-ante cash transfers individually showed on average 7.5% more reduction in food insecurity than the reactive government scenario. Improved early warning systems showed on average - over all climate scenarios- an increased reduction of 4.5%. It should be kept in mind that ADOPT does not consider (illicit) coping activities in the face of droughts such as food stocking or charcoal burning. However, both of them might reduce the food security threat. Credit schemes at 2%, individually, lead to more than 8% reduction in food insecurity levels as compared to the reactive scenario; but even then, on average net food insecurity rates increase due to climate change. A proactive intervention resulted in a food insecurity rate which is 6 percent points lower than under the reactive scenario; but still showed increases in the prevalence of food insecurity under hotter and drier conditions. A prospective intervention, combining all four interventions, was able to consistently reduce the food insecurity levels over time, even under the dry-hot climate scenario. This scenario was able to counteract the increase in food insecurity, achieving a reduction of households in food shortage over time with on average 28% compared to the reactive scenario, all climate scenarios considered.

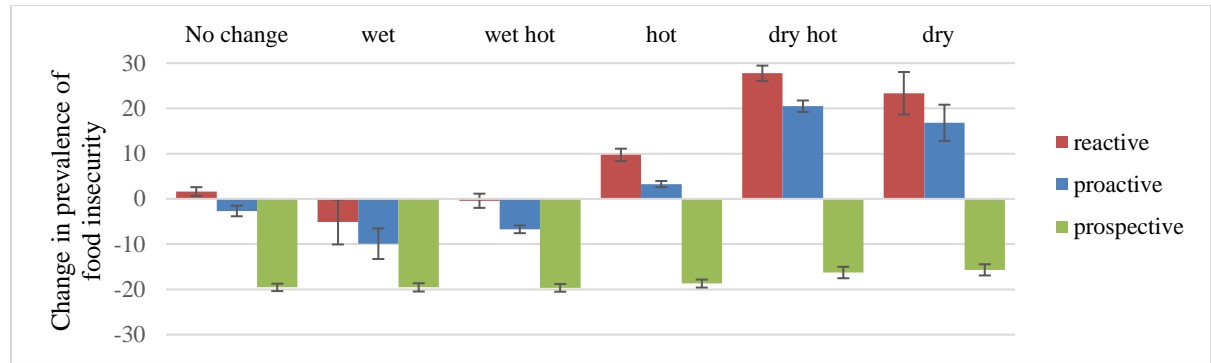

323

**Fig. 8: Absolute change (average and standard deviation introduced by sensitivity analysis - variation caused by different model initialisations and by different relative importance of the PMT factors on the decisions of households) in average share of households in food shortage of the 20 last years of scenario run, compared to the first 20 years of baseline run before "year 0", under different climate and policy intervention scenarios. ADOPT model output.**

Expressing drought impacts in average annual food aid required (in USD) can help to evaluate the effect of different climate change scenarios or different policy intervention scenarios on the drought risk of the community. These estimations are translated to USD, assuming a maize price for shortage markets, as price volatility is considered. Table 2 shows the change in aid needs compared to the no-climate change, no-top-down intervention baseline period (based on the 1980-2000 situation). When assuming no climate change, it seemed that the community is stable, only slightly increasing the share in vulnerable households. More measures were adopted as information is disseminated thought the farmer networks, but those who stay behind will face lower sell prices as markets get more stable and have a harder time accumulating assets. Under wetter conditions, reductions in drought emergency aid did reduce. However, drier, hotter climates had a detrimental effect on the food needs, with more vulnerable people crossing the food shortage threshold.

Under the no climate change scenario, each of the four policy interventions did cause a reduction in aid needs, with credit schemes having the largest effect. Under wetter conditions, they also increased the reduction of aid needs compared to the reactive scenario. However, no individual measure, was able to offset the effect of hotter and drier climate conditions. Even under a proactive intervention, there would still be an increase in aid needs under such climate conditions. Only under the prospective intervention scenario, a decrease in aid needs was visible under all possible climate change scenarios.

**Table 2: Change in aid needs (%) in 2030-2050 compared to 1980-2000 (average and standard deviation introduced by sensitivity analysis - variation caused by different model initialisations and by different relative importance of the PMT factors on the decisions of households) under different climate and policy intervention scenarios. ADOPT model output.**

| | No change | Wet | Wet Hot | Hot | Dry Hot | Dry |
|---|---|---|---|---|---|---|
| *Reactive scenario* | 4 (+-4)% | -29(+-20)% | -11(-+6)% | 37(+-6)% | 117(+-6)% | 94(+-24)% |
| *Ex ante cash transfer* | -2(+-4)% | -31(+-15)% | -20(+-5)% | 24(+-5)% | 92(+-3)% | 76(+-17)% |
| *Early warning system* | -6(+-6)% | -42(+-18)% | -24(+-6)% | 25(+-5)% | 109(+-8)% | 86(+-25)% |
| *Extension services* | -20(+-7)% | -49(+-17)% | -33(+-6)% | 15(+-4)% | 96(+-9)% | 71(+-15)% |
| *Credit at 2% rate* | -24(+-10)% | -50(+-18)% | -33(+-8)% | 10(+-12)% | 86(+-12)% | 62(+-28)% |
| *Proactive scenario* | -15(+-6)% | -48(+-12)% | -37(+-3)% | 13(+-5)% | 73(+-6)% | 58(+-17)% |
| *Prospective scenario* | -80(+-1)% | -81(+-1)% | -82%(+-1) | -78(+-2)% | -68(+-3)% | -66(+-4)% |

**5.      Discussion**

**5.1      The effect of early warning, extension services, ex-ante transfers and low interest rates**

Under a reactive strategy ("no intervention") and assuming no climate change, a slow but steady adoption of mulch, Fanya Juu, shallow well and irrigation practices is estimated. This is a result of an ever increasing information diffusion through the farmer networks and existing extension services, as also found in Hartwich et al., 2008a; van Duinen et al., 2016a; Villanueva et al., 2016; Wossen et al., 2013. Yet, multiple smallholder households still suffer from the effects of droughts, indicated by the elevated food insecurity rates and poverty rates. While some can break the cycle of drought and subsequent income losses, others are trapped by financial or other barriers and end up in poverty and recurring food insecurity. This is also found by e.g., Enfors & Gordon, (2008); Mango et al., (2009); Mosberg & Eriksen, (2015); Sherwood, (2013). In the reactive scenario, it is clear that adaptation intention is limited by factors such as a low risk perception, high (initial) adaptation costs, a limited knowledge of the adaptation efficacy or a low self-efficacy. Some of these barriers are alleviated through the different government interventions.

As compared to this reactive scenario an increased rate of adoption is observed for all policy interventions. This translates into a comparatively lower drought risk (expressed by the indicators: community poverty rate, food security and aid needs). While initially extension services have the largest effect on the adoption of on-farm drought adaptation measures, over time access to credit results in the highest adoption rates and is also estimated to decrease emergency aid the most. The former, alleviating the knowledge (self-efficacy) barrier, increases adoption under no climate change with 27% as compared to no intervention. It is indeed widely recognized as an innovation diffusion tool in different contexts (e.g., Aker, 2011; Hartwich et al., 2008b; Wossen et al., 2013). The latter, alleviation the financial (adaptation costs) barrier, increases adoption under no climate change with 30% as compared to no intervention. It is also found to be an effective policy to reduce poverty in Ghana by Wossen and Berger (Wossen & Berger, 2015). Ex-ante cash transfers also tackle the financial barrier but less effectively (the cash sum is small and fixed – only significant for less wealthy households), increasing adoption under no climate change with 25% as compared to no intervention. This matches empirical evidence on the positive effects of ex-ante cash transfers (Asfaw et al., 2017; Davis et al., 2016; Pople et al., 2021). However, ADOPT model estimations might be an underestimation as the model does not account for many preparedness strategies of households such as stocking up food while the price is still low, fallowing land to reduce farm expenses, or searching for other sources of income (Khisa & Oteng, 2014). Seasonal early warning systems, which raise awareness of upcoming droughts, increase the adoption of measures with 22% as compared to no intervention. Early warnings have a stronger effect on the adoption of mulching or Fanya Juu (cheaper measures, lower financial barrier) than on drip irrigation. Clearly, the positive effect of the interventions on household resilience varies, which is confirmed by the empirical findings of Wens et al. 2021.

The proactive government scenario, "preparing for drought disasters" by improving early warning systems and supporting ex-ante cash transfers, has a larger effect on drought risk. However, this effect is not as much as the sum of the effect of the two interventions. In contrast, the prospective government scenario "mitigating drought disasters" by combining all four interventions, alleviates multiple barriers to adoption at once. This creates a

significant, non-linear increase in adoption, matching the significant positive correlation between the preferences
for extension, credit, early warning in Wens et al. 2021. Consequently, this scenario results in a clear growth in
resilience of the farm households, shown in more stable income, lower poverty rates and less food insecurity.

## 5.2 The robustness of drought risk reduction interventions under climate change

Climate change influences the effectivity of the measures as well as farm households' experience with droughts.
Under all climate change scenarios, a lower adoption of adaptation measures compared to the "no climate change"
assumption is observed. This could be explained by the fact that the perceived need to adapt is lower under wet
conditions and the financial strength to adapt is lower under dry or hot conditions. This highlights two different
barriers to adoption: risk appraisal lowers when the occurrence of drought impacts is less frequent, while coping
appraisal lowers due to experiencing more drought impacts. This link between drought experiences, poverty and
adaptation was also found in other studies (e.g., Gebrehiwot & van der Veen, 2015; Holden, 2015; Makoti &
Waswa, 2015; Mude et al., 2007; Oluoko-Odingo, 2011; Winsen et al., 2016)
While their effect on the adoption rates seems rather small, the diverse climate change scenarios have a distinctly
different effect on the evolution of drought risk in the rural communities. Due to the adaptation choices of the
farm households, average maize harvests are estimated to slightly increase under the "no climate change"
scenario. A major increase is estimated under wet and wet-hot conditions where both increased adoption and
better maize producing weather conditions play a role. Under hot, dry and dry hot conditions, the average
household harvests are estimated to decrease (also found in Wamari et al., 2007). Increases in median and mean
assets (household wealth) are estimated slightly increase under the no climate change scenario. In this case,
adaptation efforts are able to reducing the drought disaster risk. Drier climates might lead to decreases in median
and mean assets, if farm households are not supported through top-down interventions, Hotter climates are
estimated to result decreased median but increased average assets of the households. In this case, adaptation rates
are not high enough to avoid increasing drought risk for the median households.
The proactive government scenario is estimated to level poverty and food security under hotter or drier climate
change scenarios. The prospective government scenario is the only scenario estimated to reduce emergency aid
under all possible future climates. However, it should be noted that it takes one to two decades to make a
significant difference between the reactive stance and prospective intervention plan. In other words: with climate
change effects already visible through an increased frequency of drought disasters, and more to be expected within
the following 10-20 years, prospective intervention should be started now in order to be benefit from the increased
resilience in time under any of the evaluated futures.

## 5.3 ADOPT as a dynamic drought risk adaptation model

In the past decade, the use of ABMs in *ex-post* and *ex-ante* evaluations of agricultural policies and agricultural
climate mitigation has been progressively increasing (Huber et al., 2018; Kremmydas et al., 2018). A pioneer in
agricultural ABM is Berger (2001) who couples economic and hydrologic components into a spatial multi-agent
system. This is followed more recently by for example Berger and Troost (2011), Van Oel and Van Der Veen
(2011), Mehryar et al. (2019) and Zagaria et al. (2021). The socio-hydrological, agent-based ADOPT model
follows this trend in that it fully couples a biophysical model—AquacropOS—and a social decision model—
simulating adaptation decisions using behavioural theories—through both impact and adaptation interactions.
The initial ADOPT model setup was created through interviews with stakeholders (Wens et al. 2020), and the
adaptive behaviour is based on both existing economic – psychological theory and on empirical household data
(Wens et al. 2021). The assumption of heterogeneous, bounded rational behaviour is addressed yet only by a few
risk studies (e.g. Van Duinen et al. 2015, 2016; Hailegiorgis et al. 2018, Keshavarz and Karami 2016, and Pouladi
et al. 2019). These studies have implemented empirically supported and complex behavioural theories in ABMs
similarly to ADOPT (Schrieks et al. 2021; Jager, 2021; Taberna et al., 2020; Waldman et al., 2020).
ADOPT differs from these models, however, through its specific aim to evaluate households and community
drought disaster risk beyond the number of measures adopted, crop yield, or water use. Rarely (except e.g., Dobbie
et al 2018) do innovation diffusion ABM use socio-economic metrics to evaluate drought impacts over time –
while such risk proxies are of great social relevance. As such, ADOPT evaluates the heterogeneous changes in
drought risk for farm households, influenced by potential top-down drought disaster risk reduction (DRR)
interventions. It does so through simulating their influence on individual bottom-up drought adaptation decisions
by these farm households and their effect on socio-economic proxies for drought risk (poverty rate, food security
and aid needs). To our knowledge, this is rather novel in the field of DRR and drought risk assessments.
**5.4    Uncertainties in ADOPT and limitations in investigated measures and interventions**
While yield data has been validated over the historical period (Wens et al. 2020), the model output cannot be
used as a predicting tool. This would require more extensive validations for which, currently, data is not available.
Such data would include longitudinal information on household vulnerability and adaptation choices from areas
where certain policies are being implemented, or detailed data on aid needs for the case study area. The past
average poverty and food insecurity rates matched observations (Wens et al. 2020). However, absolute amounts
of emergency aid needs are sensitive to the averages and fluctuations of household assets which proved harder to
verify. Besides, poverty and food insecurity depend also on external, food or labour market and other influences
which might change towards the future. Moreover, the simulated climate scenarios are not entirely realistic
(because variability changes are ignored and because the synthetic future data is created based on statistics rather
than physical climate and weather system changes). Moreover, the East African Climate Paradox (Funk et al.,
2021) creates its own set of challenges predicting future weather conditions in the study area.
Unavoidably, multiple possible smallholder adaptation measures are omitted in this study: many more agricultural
water management measures, agronomic actions, and other options under the umbrella of climate-smart
agriculture, exist. Besides, only four different policy interventions are evaluated while various other exists. Costs
of these top-down interventions are unknown, making cost-benefit estimates regarding drought risk reduction
strategies not possible for this study. Studying additional measures or interventions is be possible using the
ADOPT model, but requires (the collection of) more data for parametrization and calibration.
Another future improvement to the model could be to directly sample the empirical household survey data (Wens
et al 2020) to create a synthetic agent set. Now, the creation of agents (households) with different characteristics
is drawn from distribution functions based on frequencies in the empirical data. Such one-to-one data-driven
approach is similar to microsimulation and gaining popularity among ABMs (Hassan et al 2010). Lastly, the
model application does assume no shifts in the processes underlying weather and human decision making: both
the synthetic future weather situation and the decision making processes are based on past observations. To avoid
the effect of systemic changes and black swan effect, only 30 "future" years are modelled.
Because the model setup could not be fully validated, and scenarios do not provide a complete overview of all
possibilities, this study does not claim to provide a prediction of the future for south-eastern Kenya. However,
ADOPT is meant to – rather than forecast drought impact -  increase understanding of the differentiated effect of
adaptation policies: the relative differences in the risk indicators are informative for the comparison of these top-
down interventions under different changes in temperature and precipitation. This study showcases the application
of ADOPT as a decision support tool. It evaluates the robustness of a few, dedicatedly chosen policy interventions
on farm household drought risk under climate scenarios that are deemed to be relevant for the specific area. Future
research can use ADOPT to study the differentiated effect of these interventions on different types of households,
in order to tailor strategies and target the right beneficiaries of government interventions. .
**6.      Conclusion**
Top-down interventions, providing drought and adaptation information as well as supporting the capacity to act
on the basis of this information, are needed to increase the resilience of smallholder farmers to current and future
drought risk. However, to which extent these interventions will steer farmers' intention to adopt drought
adaptation measures, hence how effective they are in reducing the farm household drought risk, often remains
unknown. In this study, the agent-based drought risk adaptation model ADOPT is applied to evaluate the effect
of potential future scenarios regarding climate change and policy interventions on agricultural drought risk in
south-eastern Kenya. The smallholder farmers in this region face barriers to adopt drought adaptation measures
such as mulching, Fanya Juu terraces, shallow wells, and drip irrigation, to stabilize production and income.
ADOPT simulates their adaptive behaviour, influenced by drought occurrences under changing climate
conditions. Adaptive behaviour is also influenced by top-down (non-)government drought risk reduction
interventions such as the introduction of ex-ante cash transfers, affordable credit schemes, improved early warning
systems and tailored extension services.  We demonstrate that the investigated interventions  all increase the
uptake of adaptation measures as compared to the reactive scenario under no climate change (business-as-usual).
Extension services (+27% uptake) multiply adaptation knowledge and thus increase self-efficacy among the
smallholders, which raises the adoption of less popular drought adaptation measures. Accessible credit schemes
(+30% uptake), alleviating a financial barrier, are effective especially for more expensive drought adaptation
measures. Early warning systems (+22% uptake), creating risk awareness, are more effective in climate scenarios
with less frequent drought. Ex-ante cash transfers (+25% uptake) allow the least endowed households to climb
out of the poverty trap by adopting low-cost drought adaptation measures and thus reducing future shocks. The
effect of climate change on the adoption of adaptation measures is limited.
Moreover, this study proves that alleviating only one barrier to adoption has a limited result on the drought risk
of the farm households. Under the pro-active scenario (+40% uptake), combining early warning with ex-ante cash
transfers, smallholder farmers are better supported to adopt drought adaptation measures and to create, on average,
more wealth. However the effect of climate change on farm households risk differs significant under this proactive
scenario. While for wetter conditions, this scenario is able to increase food security and reduce poverty, this is
not sufficient to diminish the need for external food aid under every evaluated climate scenario. Only by
combining all four interventions (+139% uptake), a strong increase in the adoption of measures is estimated.
Simultaneously increasing risk perception, reducing investment costs, and elevating self-efficacy, creates
nonlinear synergies. Under such prospective government approach, ADOPT implies significantly reduced food
insecurity, decreased poverty levels, and drastically lower drought emergency aid needs after 10 to 20 years,
under all investigated climate change scenarios.
This study suggests that, in order to reach the current targets of the Sendai Framework for Disaster Risk, which
aims at building a culture of resilience, and to achieve Sustainable Development Goals "zero hunger",
"sustainable water management" and "climate resilience", a holistic approach is needed. While we present a
proof-of-concept rather than predictive model, the results improve the understanding of future agricultural
drought disaster risk under socio-economic, policy and climate trends. We provide evidence that agent-based
models such as ADOPT can serve as decision support tools to tailor drought risk reduction interventions under
uncertain future climate conditions: More research into the heterogeneous effect of the investigated top-down
interventions on households' adaptation decisions and drought risk can provide information for the effective and
efficient tailoring of the policy interventions. However, from this study, it is clear that multiple interventions -
both (risk and adaptation) information provision and the creation of action perspective - should be combined to
build a sustainable future for smallholder farmers in Kenya's drylands.

 **Appendices**

**Appendix A: Description of the ADOPT model following the ODD+D protocol** (Laatabi et al., 2018; Müller et al., 2013)**:**
**I. Overview**
**I.i Purpose**
**What is the purpose of the model?**
The purpose of ADOPT is to improve agricultural drought disaster risk assessments by including the complex
adaptive behaviour of smallholder farmers. The ADOPT model simulates the welfare (poverty level, food security
& aid needs) of smallholder farm households over time as a function of climate effects on agricultural production,
mitigated by implemented adaptation measures, and simulates the adoption of such measures as a function of
economic, social and psychological household characteristics. Understanding the two-way feedback between
households' adaptation decisions and maize yield losses over time can help optimize drought impact estimations
under climate and policy changes. ADOPT can be used to evaluate the adoption rate of adaptation measures under
different climate and policy scenarios hence contrast their effect on the drought disaster risk – approximated by
food security and welfare - of smallholder farmers.
**For whom is the model designed?**
The ADOPT model can allow scientists to increase their understanding of the socio-hydrological reality of
drought disaster risk and drought adaptation in a smallholder farming context. It can also help decision makers to
design drought policies that target specific farm household and evaluate the effect of these policies on their
drought vulnerability.
**I.ii Entities, state variables, and scales**
**What kinds of entities are in the model?**
The agents in ADOPT are individual farm households that have a farm of varying size and potentially an off-farm
income source. Two other entities exist: the crop land (multiple fields) that yields maize production and is owned
by the farm households, and the market (one) where maize is sold and bought.
**By what attributes are these entities characterized?**
Farm households (see UML, figure A.1) have a farm – characterised by its farm size and the adaptation measures
implemented on it-. They also have a family size, a household head (male/female) with a certain age and education
level, financial assets (wealth, expressed in USD), off-farm employment, and farm, food and other expenses.
Household heads have a memory regarding past drought impacts, have a perception about their own capacity,
and, in varying degrees, have information about potential adaptation measures.
Crop land (farms) (see UML, figure A.1), belonging to households, produce maize under changing weather
conditions, influenced by potential adaptation measures affecting water management conditions. The market (see
UML, figure A.1) is influenced by local production and consumption, which results in a variable maize price
depending on the balance between supply and demand. In the presented case study, we consider relatively isolated
areas, less subjected to globalized market systems: maize price is variable following the total amount of locally
produced maize to replicate the observed price volatility (with minimum and maximum prices derived from
FEWSnet) during years of reduced production.

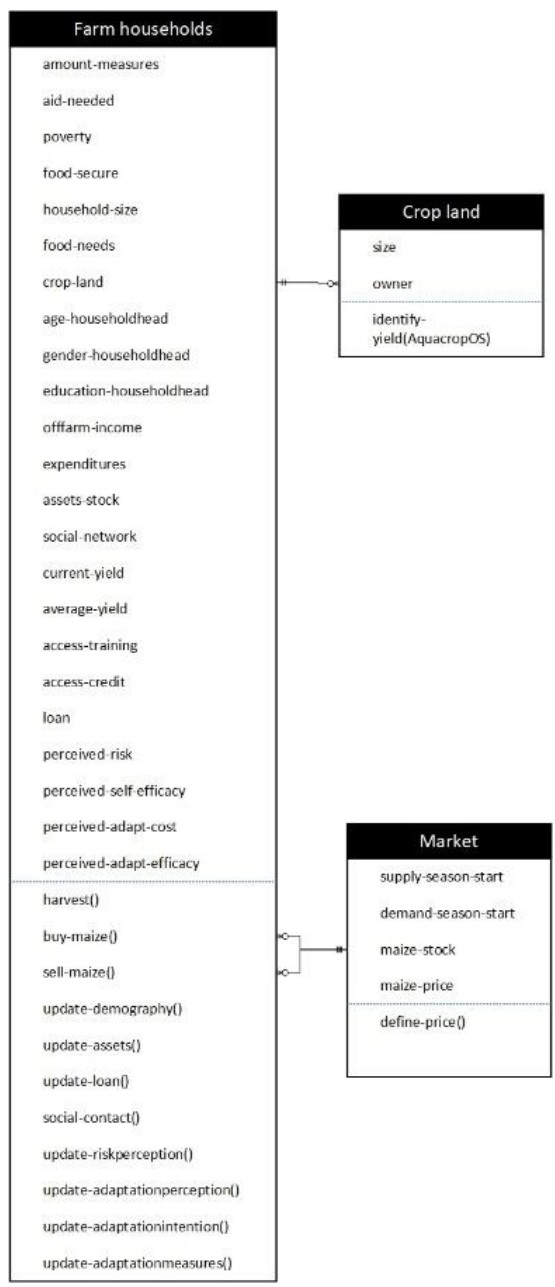


*Figure A1. UML diagram*

**What are the exogenous factors / drivers of the model?**

Two exogenous factors influence the farm household systems: daily weather (influenced by gradual climate change) and drought disaster risk reduction policies (top-down policy interventions supporting smallholder farmers). The first factor might alter the frequency and severity of droughts – which may lead to failed crop yields, while the latter affects the knowledge, access to credit, and risk perception of households who are recipient of the policies.

**How is space included in the model?**

ADOPT runs on the scale of farm fields (size adjusted to the case study area). On this field scale, agricultural water management decisions (adaptation) interact with rainfall variability (drought hazard). However, spatially-explicit fields are used only in the initialisation phase so neighbouring farms can be identified but does not play any further role: space is only represented in a spatially-implicit way, all farms (crop land) receive the same amount of rain and sun, have the same soil type with a similar slope and differ only in their farm size and management applied.

**What are the temporal resolution and extent of the model?**

One time step of ADOPT represents one year. The crop model part runs on a daily basis, producing maize crop yield in every cropping season, but decisions by the farm households to eventually adopt new adaptation measures are only made once a year. Each year, the poverty status, food security situation, and potential food aid needs of all farm households are evaluated. The model runs 30 years historical baseline (+ 10 initialisation years) and 30 scenario years.

### I.iii Process overview and scheduling

**What entity does what, and in what order?**

Every year, farm income of the households is updated with the maize harvest sold at the current market price (see centre of the flowchart in Fig. A.2). This harvest depends on the farm size of the household, the maize yields (defined by AquacropOS) which may be affected by a drought potentially mitigated by implemented drought adaptation measures, and on the food needs of the own household (subsistence is prioritized over selling; household members can die or be born (stochastically determined, based on birth and mortality rates in the study area). This farm income, together with a potential (fixed) off farm income, and with farm-size-dependent farm expenses, family-size-dependent household expenses, and potentially extra food expenses (if the own production was not sufficient to fulfil household food needs), alters the assets of the farm household. The farm household's memory of drought impacts (risk perception) is updated, and they interact (in random order) with their network of neighbours exchanging information on adaptation measures.

Once a year, the household head decides whether they want to adopt a new drought adaptation measure. They make this decision based on their memory of past drought impacts, their perception of the adaptation costs, the knowledge on adaptation measures through their networks and training, and their perception of their own capacity.

The adoption of a new measure changes the farm management of those farmers, directly changes their wealth
(implementation costs) and the farm expenses for the following years (maintenance costs), and influences crop
yield and crop vulnerability to drought – thus potential farm income - during the following years.

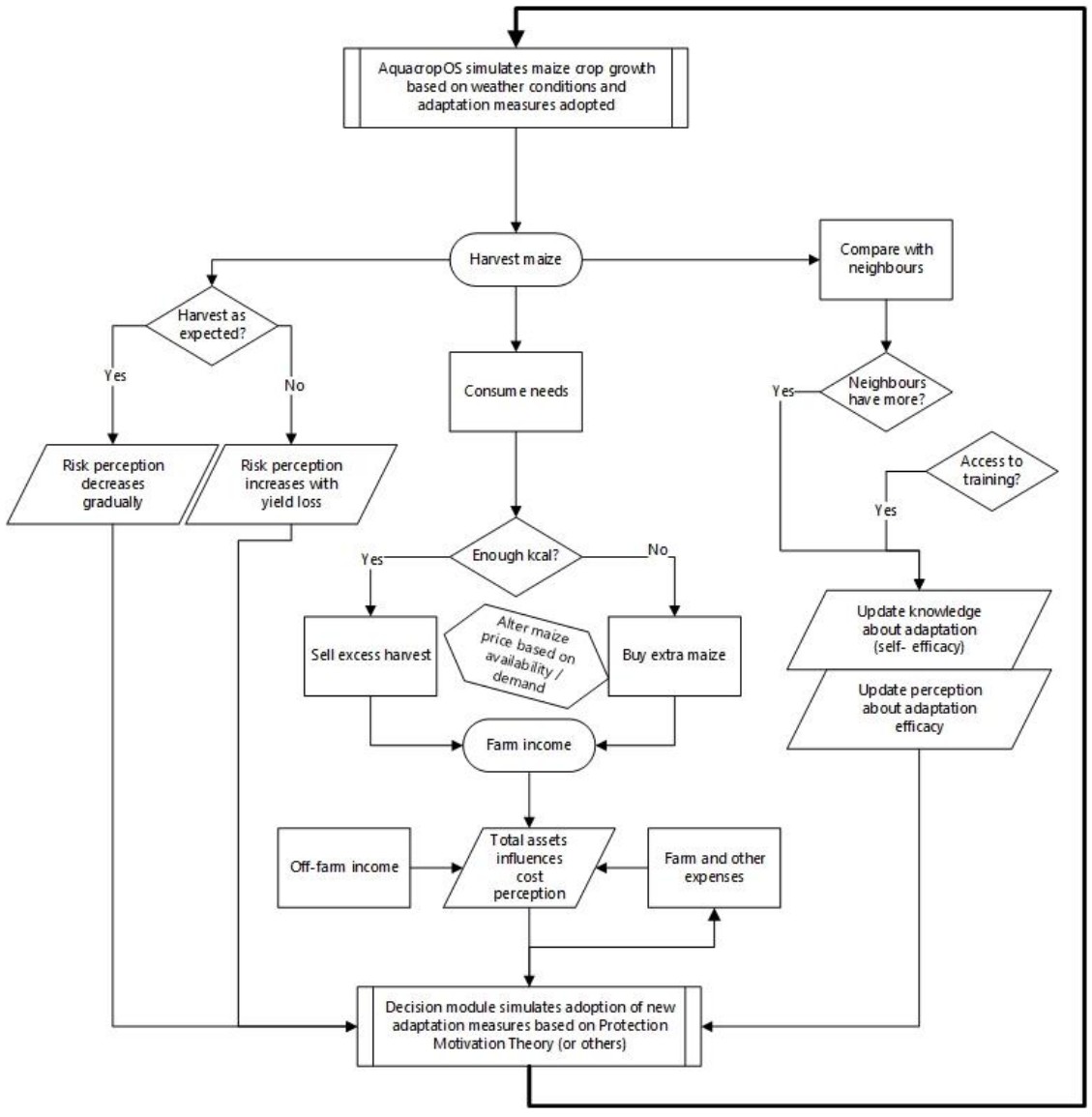

**Fig.**

**Figure A2: Flowchart showing process overview**

 **II. Design Concepts**

    **II.i Theoretical and Empirical Background**

**Which general concepts, theories or hypotheses are underlying the model's design at the system level or at**
**the level(s) of the sub-model(s) ?**
The multi-disciplinary modelling approach of ADOPT is rooted in socio-hydrology (Sivapalan et al., 2012),
where the human system both influences and adapts to the changing physical environment (in this case agricultural
drought), and applies an agent-based approach to deal with heterogeneity in adaptive behaviour of smallholder
households.
The setup / design of the model (the drought disaster risk system) is a result of participatory concept mapping
with researchers and students of SEKU University, technical advisors of Kitui County Department of Water,
Agriculture, Livestock and Fishing, experts from SASOL foundation, and five pilot households that have example
farms for agricultural extension. This information informed the decision context of ADOPT.
**On what assumptions is/are the agents' decision model(s) based?**
In the first design of ADOPT, three adaptive behaviour scenarios were analysed, with increasing complexity. A
'business as usual' scenario with no changing drought adaptation measures was tested, characterizing the 'fixed
adaptation' approach. The conventional Expected Utility Theory (von Neumann and Morgenstern, 1944)
represents the widely-used economist assessment of choice under risk and uncertainty. Simulating bounded
rational rather than economic rational adaptation decisions, the Protection Motivation Theory (Rogers, 1983) is
used as a way to include psychological factors in the heterogeneous adaptive behaviour of smallholders.
Indeed, it is often stated that households' adaptive behaviour is bounded rational and embedded in the economic,
technological, social, and climatic context of the farmer (Adger, 2006). Knowing the risk is not enough to adapt;
farmers should also believe the adaptation measure will be effective, be convinced that they have the ability to
implement the measure, and be able to reasonably pay the costs (van Duinen et al., 2015b). Financial or knowledge
constraints may limit economic rational decisions. Also age, gender and education – intrinsic factors - can play a
role (Burton, 2014). The perceived ability to do something (Coping Appraisal) influences the decision making
process(Eiser et al., 2012). This coping appraisal can be subject to intrinsic factors such as education level, sources
of income, farm size, family size, gender, confidence and beliefs, risk-aversion, and age (Le Dang et al., 2014;
Okumu, 2013; Shikuku et al., 2017; Zhang et al., 2019) .
In order to understand the observed adaptive behaviour of smallholder households, it is critical to incorporate
such social-economic factors in the decision-making framework of drought adaptation models (Bryan et al., 2009,
2013; Deressa et al., 2009; Gbetibouo, 2009; Gebrehiwot & van der Veen, 2015; Keshavarz & Karami, 2016;
Lalani et al., 2016; Mandleni & Anim, 2011; O'BRIEN et al., 2007; Rezaei et al., 2017; Singh & Chudasama,
2017; van Duinen et al., 2015b, 2015a, 2016; Wheeler et al., 2013). After we had promising results running
ADOPT with the bounded rational scenario, it is assumed that farmers show a bounded rationality in the further
application of ADOPT.

**Why is a/are certain decision model(s) chosen?**

Analysis of the past and intended behaviour of farm households in the region provided support for the choice of theory, but also showed the need to include network influencing risk perception and capacity of the households. Besides helping to parameterize the model, it also helped to calibrate the influence of the different factors affecting the decision making process of the farm household. Showing the effect of different assumptions about decision making in the first exploration of ADOPT (M. Wens et al., 2020), and with empiric evidence on the adaptive behaviour (M. L. K. Wens et al., 2021), the decision rules in ADOPT are assumed be a good enough representation of the decision making process regarding drought adaptation.

**If the model / a sub-model (e.g., the decision model) is based on empirical data, where does the data come from?**

ADOPT is designed/initialised with data from existing longitudinal household surveys (Tegemeo Institute, 2000, 2004, 2007, 2010) and from a fuzzy cognitive map of key informants, and parameterized/partially calibrated with data from a semi-structured household questionnaire among 260 smallholder farmers Survey reports can be found here:
- https://research.vu.nl/en/publications/survey-report-kitui-kenya-expert-evaluation-of-model-setup-and-pr
- https://research.vu.nl/en/publications/survey-report-kitui-kenya-results-of-a-questionaire-regardings-us

**At which level of aggregation were the data available?**

Data from the surveys are available on individual household level.

### II.ii Individual Decision Making

**What are the subjects and objects of decision-making? On which level of aggregation is decision-making modelled?**

In ADOPT, individual farm households make individual adaptation decisions about their farm water management (in the case study in Kenya: mulching, Fanya Juu terraces, drip irrigation or shallow well) to reduce their production vulnerability to droughts. There are no multiple levels of decision making included.

**What is the basic rationality behind agents' decision-making in the model? Do agents pursue an explicit objective or have other success criteria?**

Farmers generally try to reduce their drought disaster risk (achieve food security, evade poverty and avoid needing emergency aid) and thus try to maximise crop yields (diminish yield reduction under water-limited conditions) given the capacity they have to adopt adaptation measures.

**How do agents make their decisions?**

The Protection Motivation Theory (Maddux & Rogers, 1983) (see II.i) is used to explain the decision making process of the households. PMT consists of two underlying cognitive mediating processes that cause individuals to adopt protective behaviours when faced with a hazard (Floyd et al., 2000): It suggests that the intention to

protect (in this study, the farmers' intention to adopt a new adaptation measure) is motivated by a persons' risk
appraisal and the perceived options to cope with risks. The former depends on, for example, farmers' risk
perception, on their own experiences with drought disasters and memory thereof, and on experiences of risk
events in their social networks. The latter is related to different factors such as perceived self-efficacy (i.e. assets
and sources of income, education level, and family size), adaptation efficacy (land size, adaptation measure
characteristics) and adaptation costs (expenses in relation to their income) (Gebrehiwot & van der Veen, 2015;
Keshavarz & Karami, 2016; van Duinen et al., 2015, 2016a). Households do not have any other objective or
success criteria. A detailed description of how PMT is modelled – including the sensitivity analysis regarding the
relative weights of the PMT factors - can be found in Wens et al. (2019): In ADOPT, farm households develop
an intention to adapt (protect) for each potential adaptation measure (m) which changes every year (t). If a
household has the financial capacity to pay for a considered measure (Stefanovi, 2015), the intention to adapt is
translated into the likelihood the household will adopt this measure in the following years. (This can be influenced
by having access to credit.) The actual adoption is stochastically derived from this likelihood to adopt a measure.

$$IntentionToAdapt_{t,m} = \alpha * RiskAppraisal_t + \beta * CopingAppraisal_{t,m}$$


Although Stefanovi (2015), Van Duinen et al. (2015a), and Keshavarz and Karami (2016) have found positive
relationships between the factors of PMT and observed protective behaviour, a level of uncertainty exists related
to the relative importance of risk appraisal and coping appraisal in the specific context of smallholder households'
adaptation decisions in semi-arid Kenya. Therefore, the α and β parameters were introduced as weights for the
two cognitive processes. To address the associated uncertainty, they were widely varied (α, β ϵ [0.334:0.666]) in
a sensitivity analysis.
Risk appraisal is formed by combining the perceived risk probability and perceived risk severity, shaped by
rational and emotional factors (Deressa et al., 2009, 2011; Van Duinen et al., 2015b). Whereas risk perception is
based in part on past experiences, several studies have suggested that households place greater emphasis on recent
harmful events (Gbetibouo, 2009; Rao et al., 2011; Eiser et al., 2012). To include this cognitive bias, risk appraisal
is seen as a sort of subjective, personal drought disaster memory, defined as follows (Viglione et al., 2014):

$$RiskAppraisal_t = RiskAppraisal_{t-1} + (Drought_t * Damage_t) - 0.125 * RiskAppraisal_{t-1} \text{ with } Damage_t = 1 - \exp(-harvestloss_t)$$


The drought occurrence in year t is a binary value with a value of 1 if the SPEI-3 value falls below −1. The disaster
damage of a household is related to their harvest loss during the drought year, which is defined as the difference
between their current and average harvest over the last 10 years.
Coping Appraisal represents a households' subjective "ability to act to the costs of a drought adaptation measures,
given the adaptation measures' efficiency in reducing risk" (Stefanovi, 2015; Van Duinen et al., 2015a). It is a
combination of the households' self-efficacy, adaptation efficacy of the measure, and its adaptation costs:

$$CopingAppraisal_{t,m} = \gamma * SelfEfficacy_t + \delta * AdaptationEfficacy_{t,m}$$
$$+ \varepsilon * (1 - Adaptationcosts_t)$$


Although Stefanovi (2015), Van Duinen et al. (2015b), and Keshavarz and Karami (2016) quantified the
relationships between the factors driving the subjective coping appraisal of individuals, a level of uncertainty
remains related to the relative importance of these drivers in the context of smallholder households' adaptation
decisions in semi-arid Kenya. Therefore, weights ($\gamma$, $\delta$, $\varepsilon \in [0.25:0.50]$) were introduced and varied in a sensitivity
analysis using different ADOPT model runs.
The Adaptation Costs of the possible measures are expressed in terms of a percentage of the households' assets.
The Adaptation Efficacy is calculated as the percentage of yield gain per measures compared to the current yield.
This can be influenced by access to extension services (which gives an objective yield gain based on future climate
rather than an estimate based on current practices of neighbours)
Self-efficacy is assumed to be influenced by education level (capacity), household size (labour force), age and
gender; all social factors found to influence risk aversion and adaptation decision (Oremo, 2013; Charles et al.,
2014; Tongruksawattana, 2014; Muriu et al., 2017).
**Do the agents adapt their behaviour to changing endogenous and exogenous state variables? And if yes,**
**how?**
Exogenous factors influencing adaptation decisions in ADOPT include the climate and the policy context in which
households exists. Drought (a feature of the climate context) induced crop losses steer a households' perception
of the drought disaster risks they face (Risk Appraisal). For example, experiences of historical droughts or
receiving early warnings about upcoming drought affects individuals' evaluation of drought disaster risk, leading
to a personal drought disaster risk judgement (e.g. Keshavarz et al., 2014; Singh & Chudasama, 2017). Besides,
access to extension services (a feature of the climate context) can have profound effect on whether or not
individuals take proactive action (Kitinya et al., 2012; Shikuku et al., 2017). Endogenous factors, as explained
above, include age, household size, education level, maize yield variability and assets (and the potential access to
credit market).
**Do spatial aspects play a role in the decision process?**
Farmer networks (connections with neighbours) exist, and information is passed through this social network.
**Do temporal aspects play a role in the decision process?**
Yes, risk memory is based on the crop yield variability of the accumulated past years and gives farm households
an expectation about the upcoming crop yield.
**Do social norms or cultural values play a role in the decision-making process?**
No (only implicitly included, see II.ix)
**To which extent and how is uncertainty included in the agents' decision rules ?**
No
**II.iii Learning**
**Is individual learning included in the decision process? How do individuals change their decision rules over**
**time as consequence of their experience?**
Decision rules follow the PMT and are thus fixed, but some rules differ among type of households. Households
that do not regularly receive extension services, are limited to only implement measures that their neighbours
have installed as they are not aware of the existence of others. Besides, farmers who receive training will form
their perception about the adaptation efficacy in a more objective way (as they have knowledge of average yield
results under the adaptation measures while other farmers estimate this based on yield of their peers with such
measure).
**Is collective learning implemented in the model?**
No
**II.iv Individual Sensing**
**What endogenous and exogenous state variables are individuals assumed to sense and consider in their**
**decisions? Is the sensing process erroneous?**
Households are aware of their assets, past yields, income sources and their stability, and household food needs
(Fig. A1). Following the socio-hydrologic setup of the model, households with bounded rational behaviour are
embedded in and interact with their social and natural environment. Changes in rainfall patterns during the
growing season will change households' risk perception through fluctuations in crop yield; drought memory will
influence the adaptive behaviour of these households. Besides, there is a diffusion of technology due to
interactions and knowledge exchanges among farm households as discussed above.
**What state variables of which other individuals can an individual perceive?**
Households know their own but also their neighbours' current yields and management practices. They make
assumptions about the adaptation efficacy based on this.
**What is the spatial scale of sensing?**
Individual sensing happens on household level, but also through the individual social network that the farmers
have, containing 3 to 30 other farmers.

**Are the mechanisms by which agents obtain information modelled explicitly, or are individuals simply assumed to know these variables?**

Households can get information about early warnings and through extension training. Households also have a simulated information transfer moment with the farmers in their neighbourhood to exchange information on risk and yields.

**Are the costs for cognition and the costs for gathering information explicitly included in the model?**

No

### II.v Individual Prediction

**Which data uses the agent to predict future conditions?**

By extrapolating from historical yield experiences, farmers have expectations about their maize yield every year. If an early warning system is in place, farmers know about upcoming droughts that can influence their crop yield.

**What internal models are agents assumed to use to estimate future conditions or consequences of their decisions?**

Households receiving extension services have knowledge about the average (future) yield gain of adopting a new adaptation measure, which will influence their coping appraisal.

**Might agents be erroneous in the prediction process, and how is it implemented?**

Households without this access to training will predict the yield gain based on the extra yield of their neighbours who have already adopted the considered adaptation measure.

### II.vi Interaction

**Are interactions among agents and entities assumed as direct or indirect?**

In ADOPT, households interact with their neighbours, shaping risk awareness and response attitude (Nkatha, 2017; Okumu, 2013; van Duinen et al., 2016). Such networks can enhance social learning and knowledge spill over, which influences people's adaptation intention and choice of specific measures (Below et al., 2010; Tongruksawattana, 2014). Smallholder households learn from the other households in their social network about the implementation and benefits of drought adaptation measure through neighbouring households' (Below et al 2010; Shikuku 2017). In ADOPT, exchanges with neighbours shape risk perception – the individual perception moves in the direction of the social network average – and also shape perceived adaptation effectivity. Moreover, households with no access to extension can only adopt measures already implemented by neighbours.

**On what do the interactions depend?**

Households are either more self-oriented, discussing matters with 10 neighbours, or group-oriented, sharing knowledge within a group / collective of 30 neighbouring households.

Spatial distance (neighbourhood) at initialisation is the key driver for networks; it is assumed that s(he) would
not walk more than 5km to reach people in her/his network.
**If the interactions involve communication, how are such communications represented?**
Communication is not explicitly modelled.
**If a coordination network exists, how does it affect the agent behaviour? Is the structure of the network**
**imposed or emergent?**
No coordination network exists.

**II.vii Collectives**

**Do the individuals form or belong to aggregations that affect, and are affected by, the individuals?  How**
**are collectives represented?**
No, no fixed collectives exist as the social networks the agents have, are individual in nature.

**II.viii Heterogeneity**

**Are the agents heterogeneous? If yes, which state variables and/or processes differ between the agents?**
Household agents are heterogeneous in terms of state variables (i.e. farm size, household size, assets), and differ
in access to credit market, extension services and early warning beneficiaries, changing their adaptive behaviour
(Asfaw et al., 2017; Okumu, 2013; Shikuku et al., 2017)
**Are the agents heterogeneous in their decision-making? If yes, which decision models or decision objects**
**differ between the agents?**
Okumu (2013), Shikuku (2017), among others, found that state variables such as age, beliefs. gender, education
of the household head, and the household size have significant effects on their risk attitude. These factors are
included in the model application of the Protection Motivation Theory through the self-efficacy factor.

**II.ix Stochasticity**

**What processes (including initialization) are modelled by assuming they are random or partly random?**
The likelihood to adopt a measure of a household is directly derived from the intention to adapt of the measure
with the highest intention for that household. This is stochastically transferred into an actual decision whether or
not to adopt the measure. For every time step of the simulation, a random number between 0-1 is drawn for each
household; if this is lower than their adaptation intention (also between 0-1) and the household is able to pay for
the measure, then the household adopts it. This probabilistic way of looking at adaptation intention and the
stochastic step to derive the actual decisions allow to account for non-included factors introducing uncertainty in
adaptive behaviour such as conservatism, social / cultural norms, physical health, ambitiousness etc. of the
households. Moreover, also a stochastic perturbation (multiplied with a random number with average 1 and SD
0.1) is added to the maize yield per farm as calculated through AquacropOS. This additional heterogeneity-
inducing step is done to include effects of pests and diseases on the income and food security of farming
households.
**II.x Observation**
**What data are collected from the ABM for testing, understanding and analysing it, and how and when are**
**they collected?**
The adoption of adaptation measures and their effect on the total crop production (and food stock on the market)
and individual household wealth are tracked over the simulated years.
**What key results, outputs or characteristics of the model are emerging from the individuals?**
Drought disaster risk (the annual average of impacts over the run period) - expressed in terms of average annual
poverty rate, level of food security and total emergency aid needs - is emerging from the model. They are defined
based on the socio-economic conditions of individual farm households.
**III. Details**
**II.i Implementation**
**How has the model been implemented?**
The model is coded in R, which is able to link the two sub models in Netlogo (the adaptive behaviour sub model)
and MATLAB (AquacropOS).
**Is the model accessible, and if so, where?**
No(t) yet
**III.ii Initialization**
**What is the initial state of the model world, i.e., at time t=0 of a simulation run?**
At the initial stage, households and their characteristics are randomly created based on the mean and standard
deviation (Table A1) derived from the household dataset, obtained from a survey on agricultural drought disaster
risk with smallholders in the case study area (Wens, 2019). Income off farm is linearly related to the household
size, education level and negatively related to the farm size. Food and non-food expenditures are linearly related
to the household size. Farm expenditures are linearly related to the farm size.

 **Table A1: Initialisation parameters for farm households in ADOPT**

| Parameter | Explanation of initialization parameters for farm households | Value |
|-----------|-------------------------------------------------------------|-------|
| **Age** | Age of the household head (based on Wens 2019) | 42 +- 9 |
| **Edu** | Years of education of the household head (based on Wens 2019) | 6 +- 3 |
| **Sex** | Gender of the household head (male 1, female 0) | 0.66 |
| **HH-size** | Family size of the households (people living under same roof) (Wens 2019) | 6 +- 2.5 |
| **Assets** | Household financial assets (USD) that can be spend (based on IFPRI 2012) | 80% < 100 |
| **Farm-size** | Size of the farm (in hectare) used for planting crops (Wens 2019) | 0.7 +- 0.6 |
| **Off-farm** | Income from activities not on the own farm in USD (Wens 2019) | 1200 +- 500 |
| **Food-needs** | Kilogram of maize to fulfil daily caloric intake needs, per adult | 125 |
| **Exp-farm** | Farm expenditures made by the household (USD/hectare/year) (Wens 2019) | 118 +- 146 |
| **Exp-food** | Food expenditures made by the household (USD/year) (Wens 2019) | 567 +- 655 |
| **Exp-nonf** | Other expenditures made by the household (USD/year) (Wens 2019) | 446 +- 500 |
| **Network** | Neighbouring farmers creating the social network of the farmer | 10-30 |


**Is initialization always the same, or is it allowed to vary among simulations?**

In ADOPT, multiple climate change scenarios and policy scenarios were initialised – this changed the exogeneous variables in the model. Moreover, each initialization creates another synthetic agent set based on the average household characteristics,  Besides, a sensitivity analysis is done to evaluate assumptions on the relative weights of the PMT factors (II.ii). Each combination of climate and policy scenario is run 12 times  (3 possible α; 4 possible combinations of  γ, δ, ε) to account for the endogenous variability and uncertainty.

**Are initial values chosen arbitrarily or based on data?**

The initialisation values are based on observed household data. Survey data includes a short questionnaire among employees of the Kenyan national disaster coordination units (n=10), semi-structured expert interviews (n=8) with NGOs, governmental water authorities and pioneer farmers in the Kitui district in Kenya, and an in-depth questionnaire among 250 smallholder farmers in the central Kitui. Extra information is derived from household surveys of 2000, 2004, 2007 and 2010, conducted by the Tegemeo Agricultural Policy Research Analysis (TARAA) Project of the Tegemeo Institute. Besides, the model initialization draws heavily from reports of CIAT (CIAT & World Bank, 2015), FAO (Rapsomanikis, 2010), IFPRI (Erenstein et al., 2011) and the government of Kenya (Kitui County Integrated report 2013-2017, 2017), CCAFS (CCAFS, 2015), and from research (e.g., Muhammad et al., 2010).

**III.iii Input Data**

**Does the model use input from external sources such as data files or other models to represent processes that change over time?**

The daily weather conditions from 1980-2010 (from CHIRPS and CFSR) is used as input time series; for the future climate scenarios, the same data but with temperature and/is used.

Besides, survey data on household behaviour and drought risk context are used. Raw reporting can be found in:

- Wens, M. (2019). Survey report Kitui, Kenya: Results of a questionnaire regarding subsistence farmers' drought risk and adaptation behaviour. https://research.vu.nl/ws/portalfiles/portal/98864069/MissionRapport.pdf
- Wens, M (2018) Survey report Kitui, Kenya: Expert evaluation of model setup and preparations of future fieldwork  https://research.vu.nl/ws/portalfiles/portal/98863978/MissionRapport2018.pdf

**Where does data come from? How is it collected? What is the level of available data? How is it structured?**

Data (also discussed in Wens et al. 2021) is collected in the field using a multi-method data survey approach (key informant interviews, fuzzy cognitive map, household questionnaire and choice experiment). This data is used to design the model, to validate the use of PMT, to initialise the agent set and to calibrate model outputs.

**What are the variables, entities and classes available in data? What do they represent?**

A full set of behavioural factors were evaluated through the household questionnaire, and these were linked to their actual behaviour and to their behavioural intentions, as well as to the results of the choice experiment investigating future behaviour (Wens et al. 2021). Besides, socio-economic and farm characteristics were questioned.

**How are data selected to form the agent entities? How is agent population generated and synthesized?**

As discussed above, the data is used to create a representative set of agents. Household variable means and standard deviations were used to create distribution functions and a synthetic agent set was created based on random draws from these functions. Moreover, correlation between different variables were maintained.

**What are the relationships and patterns that exist in data?**

As discussed above, relationship between household income and household head education level or farm size exist. Next to corelations between socio-economic or agricultural characteristics, correlations between psychological factors and actual or prospective adaptation decisions were investigated and used to design the behavioural module of ADOPT.

**III.iv Sub-models**
**What, in detail, are the sub-models that represent the processes listed in 'Process overview and**
**scheduling'?**
The FAO crop-water model AquacropOS (coded in MATLAB© by Tim Foster (Foster et al., 2017)) calculates
seasonal crop production, based on hydro-climatologic conditions provided by the climate data and based on the
agricultural management of the households. The agent-based model in which farming households decide on their
drought adaptation measures, is coded in Netlogo®, a language specialized in ABMs. This contains the -making-
decision module, which is a model-application of the Protection Motivation theory as explained in section II.i.
More detailed explanation about how this is done can be found in Wens et al 2020.
**How were sub models designed or chosen, and how were they parameterized and then tested?**
AquacropOS was applied parameterized and calibrated following Ngetich (2011) and Omyo (2015), who both
analysed and approved the functioning of this model to simulate maize yield under different climates in Kenya.
The decision sub-model is described above in the sections about decision-making and theoretical foundations
(II.ii). A more detailed description can be found in Wens et al 2020.
**What are the model parameters, their dimensions and reference values?**
For AquacropOS, Table A3 and A4 give an overview of the parameters that are used. For the decision-making
module, Table A2 gives an overview of the factors used.
**Table A2: Initialisation parameters for the behavioural module in ADOPT**

| Factor | Explanation of the PMT factors |
|---|---|
| **Current Yield** | Average yield of last 5 years |
| **Potential Yield** | Expected / perceived yield when adopting a new adaptation measure<br>Either based on yield of neighbours with that measure or on training info |
| **Adaptation costs** | Perception of the costs of new measures as percentage of assets |
| **Knowledge-measures** | 1 if attending trainings, else the percentage of people in network with measure |
| **Risk perception** | Drought memory, 1 if last harvest there was 0 yield, 0 if never impacted |
| **Adaptation efficacy** | Yield gain as percentage of current yield, based on potential yield |
| **Self – efficacy** | Belief in own capacity, based on gender, age, HH size and access to training |
| **Adaptive capacity** | Product of self-efficacy, adaptation efficacy and -1 * adaptation costs |
| **Adaptation intention** | Product of adaptive capacity and risk perception, 0 if one of the underlying factors is 0 or if assets are smaller than costs of measure |


**Table A3: Initialisation parameters for AquacropOS in ADOPT**

| Value | Explanation of calibration parameters for AquacropOSv6.0 maize |
|---|---|
| **60 / 80** | Curve number value under Fanya Juu bunds or under absence of such bunds |
| **06** | Bund height (m) |
| **50** | Area of surface covered by mulches (50%) |
| **0.5** | Soil evaporation adjustment factor due to effect of mulches |
| **SMbased** | Irrigation method |
| **7 / 3** | Interval irrigation in days under manual / automated irrigation |
| **40** | Soil moisture target (% of TAW below which irrigation is triggered) |
| **12** | Maximum irrigation depth (mm/day) |
| **50 / 75** | Application efficiency under manual / automated irrigation |
| **50** | Soil surface wetted by irrigation (%) |


**Table A4: Crop parameters for maize AQUACROPOS in ADOPT**
Value        Crop parameters for AquacropOS
3        : Crop Type (1 = Leafy vegetable, 2 = Root/tuber, 3 = Fruit/grain)
1        : Planting method (0 = Transplanted, 1 = Sown)
1        : Calendar Type (1 = Calendar days, 2 = Growing degree days)
0        : Convert calendar to GDD mode if inputs are given in calendar days (0 = No; 1 = Yes)
16/03     : Planting Date (dd/mm)
31/08     : Latest Harvest Date (dd/mm)
5        : Growing degree/Calendar days from sowing to emergence/transplant recovery
40       : Growing degree/Calendar days from sowing to maximum rooting
80       : Growing degree/Calendar days from sowing to senescence
90       : Growing degree/Calendar days from sowing to maturity
40       : Growing degree/Calendar days from sowing to start of yield formation
5        : Duration of flowering in growing degree/calendar days (-999 for non-fruit/grain crops)
65       : Duration of yield formation in growing degree/calendar days
3        : Growing degree day calculation method
8        : Base temperature (degC) below which growth does not progress
30       : Upper temperature (degC) above which crop development no longer increases
1        : Pollination affected by heat stress (0 = No, 1 = Yes)
35       : Maximum air temperature (degC) above which pollination begins to fail
40       : Maximum air temperature (degC) at which pollination completely fails
1        : Pollination affected by cold stress (0 = No, 1 = Yes)
10       : Minimum air temperature (degC) below which pollination begins to fail
5        : Minimum air temperature (degC) at which pollination completely fails
1        : Transpiration affected by cold temperature stress (0 = No, 1 = Yes)
12       : Minimum growing degree days (degC/day) required for full crop transpiration potential
0        : Growing degree days (degC/day) at which no crop transpiration occurs
0.3      : Minimum effective rooting depth (m)
0.8      : Maximum rooting depth (m)

| 938 | 1.3 | : Shape factor describing root expansion |
|------|--------|---|
| 939 | 0.0105 | : Maximum root water extraction at top of the root zone (m3/m3/day) |
| 940 | 0.0026 | : Maximum root water extraction at the bottom of the root zone (m3/m3/day) |
| 941 | 6.5 | : Soil surface area (cm2) covered by an individual seedling at 90% emergence |
| 942 | 37000 | : Number of plants per hectare |
| 943 | 0.89 | : Maximum canopy cover (fraction of soil cover) |
| 944 | 0.1169 | : Canopy decline coefficient (fraction per GDD/calendar day) |
| 945 | 0.2213 | : Canopy growth coefficient (fraction per GDD) |
| 946 | 1.05 | : Crop coefficient when canopy growth is complete but prior to senescence |
| 947 | 0.3 | : Decline of crop coefficient due to ageing (%/day) |
| 948 | 33.7 | : Water productivity normalized for ET0 and C02 (g/m2) |
| 949 | 100 | : Adjustment of water productivity in yield formation stage (% of WP) |
| 950 | 50 | : Crop performance under elevated atmospheric CO2 concentration (%) |
| 951 | 0.48 | : Reference harvest index |
| 952 | 0 | : Possible increase of harvest index due to water stress before flowering (%) |
| 953 | 7 | : Coefficient describing positive impact on harvest index of restricted vegetative growth during yield formation |
| 954 | 3 | : Coefficient describing negative impact on harvest index of stomatal closure during yield formation |
| 955 | 15 | : Maximum allowable increase of harvest index above reference value |
| 956 | 1 | : Crop Determinacy (0 = Indeterminant, 1 = Determinant) |
| 957 | 50 | : Excess of potential fruits |
| 958 | 0.02 | : Upper soil water depletion threshold for water stress effects on affect canopy expansion |
| 959 | 0.20 | : Upper soil water depletion threshold for water stress effects on canopy stomatal control |
| 960 | 0.69 | : Upper soil water depletion threshold for water stress effects on canopy senescence |
| 961 | 0.80 | : Upper soil water depletion threshold for water stress effects on canopy pollination |
| 962 | 0.35 | : Lower soil water depletion threshold for water stress effects on canopy expansion |
| 963 | 1 | : Lower soil water depletion threshold for water stress effects on canopy stomatal control |
| 964 | 1 | : Lower soil water depletion threshold for water stress effects on canopy senescence |
| 965 | 1 | : Lower soil water depletion threshold for water stress effects on canopy pollination |
| 966 | 1 | : Shape factor describing water stress effects on canopy expansion |
| 967 | 2.9 | : Shape factor describing water stress effects on stomatal control |
| 968 | 6 | : Shape factor describing water stress effects on canopy senescence |
| 969 | 2.7 | : Shape factor describing water stress effects on pollination |

**Appendix B: Adoption rates of adaptation measures**

**Table B1 Adoption ratio (in share of population) at run year 30 under different climate and intervention scenarios. Note that the model showed an adoption rate of 25% for mulch, 70% for Fanya Juu, 9% for well and X% for irrigation at run year 0 (start of climate change and policy scenarios) .**

| *Mulch* | **No Change** | **Wet** | **Wet Hot** | **Hot** | **Dry Hot** | **Dry** |
|---|---|---|---|---|---|---|
| *Reactive* | 50.2% | 47.8% | 45.6% | 42.1% | 35.9% | 38.5% |
| *Proactive* | 83.8% | 83.6% | 89.4% | 90.1% | 90.7% | 88.1% |
| *Prospective* | 100% | 100% | 100% | 100% | 100% | 100% |
| *Fanya Juu* | No Change | Wet | Wet Hot | Hot | Dry Hot | Dry |
| *Reactive* | 71.1% | 70.9% | 69.1% | 68.8% | 60.7% | 63.3% |
| *Proactive* | 87.2% | 88.1% | 90.7% | 90.9% | 91.9% | 90.1% |
| *Prospective* | 93.7% | 93.5% | 94.7% | 94.8% | 95.1% | 94.9% |
| *Well* | No Change | Wet | Wet Hot | Hot | Dry Hot | Dry |
| *Reactive* | 9.4% | 9.6% | 9.4% | 9.2% | 9.1% | 9.0% |
| *Proactive* | 11.7% | 12.7% | 13.4% | 12.0% | 12.1% | 11.4% |
| *Prospective* | 79.4% | 82.6% | 92.1% | 92.9% | 95.0% | 91.1% |
| *Irrigation* | No Change | Wet | Wet Hot | Hot | Dry Hot | Dry |
| *Reactive* | 3.7% | 3.7% | 3.5% | 3.4% | 3.3% | 3.4% |
| *Proactive* | 5.2% | 5.6% | 5.6% | 5.3% | 5.2% | 4.8% |
| *Prospective* | 48.7% | 59.6% | 73.3% | 75.8% | 82.0% | 71.8% |

**Table B2 Difference in adoption RATIO (in share of population) under different climate and intervention scenarios compared to the reactive government scenario under no climate change (the BAU scenario).**

| mulch | No Change | Wet | Wet Hot | Hot | Dry Hot | Dry |
|---|---|---|---|---|---|---|
| Reactive | 0 | -2.5% | -4.6% | -8.1% | -14.3% | -11.6% |
| Proactive | 33.7% | 33.4% | 39.3% | 39.9% | 40.5% | 38.0% |
| Prospective | 49.4% | 49.4% | 49.8% | 49.8% | 49.8% | 49.8% |
| EWS | 18.0% | 19.7% | 18.8% | 13.5% | -4.5% | 1.2% |
| transfer | 23.2% | 14.4 | 19.6% | 24.6% | 23.8% | 18.4% |
| Credit2 | 19.5% | 16.6% | 14.7% | 8.5% | 5.4% | 9.1% |
| training | 30.1% | 27.6% | 24.9% | 20.4% | 10.8% | 15.1% |

| Fanya Juu | NC | Wet | Wet Hot | Hot | Dry Hot | Dry |
|---|---|---|---|---|---|---|
| Reactive | 0% | -0.2% | -2% | -2.3% | -10.3% | -7.7% |
| Proactive | 16.2% | 17.0% | 19.6% | 19.8% | 20.8% | 19.1% |
| Prospective | 22.6% | 22.4% | 23.6% | 23.8% | 24.1% | 23.8% |
| EWS | 8.2% | 9.2% | 8.5% | 6.0% | -0.2% | 1.3% |
| transfer | 9.0% | 5.9% | 6.9% | 10.3% | 10.1% | 8.4% |
| Credit2 | 8.0% | 7.3% | 5.1% | 6.0% | -0.1% | 1.5% |
| training | -1.7% | -2.9% | -5.1% | -5.5% | -11.2% | -9.9% |

| Well | NC | Wet | Wet Hot | Hot | Dry Hot | Dry |
|---|---|---|---|---|---|---|
| Reactive | 0% | 0.2% | -0.1% | -0.3% | -0.4% | -0.4% |
| Proactive | 2.4% | 3.2% | 3.9% | 2.6% | 2.7% | 2.0% |
| Prospective | 69.9% | 73.2% | 82.7% | 83.4% | 85.5% | 81.6% |
| EWS | 1.7% | 2.% | 1.4% | 1.1% | -0.4% | 0.2% |
| transfer | 10.% | 1.0% | 1.1% | 0.2% | 0.4% | 0.2% |
| Credit2 | 9.4% | 9.1% | 7.4% | 6.9% | 4.2% | 5.1% |
| training | 5.2% | 5.5% | 4.4% | 3.2% | 1.5% | 1.9% |

| Irrigation | NC | Wet | Wet Hot | Hot | DRY | Dry Hot |
|---|---|---|---|---|---|---|
| Reactive | 0% | 0% | -0.1% | -0.3% | -0.4% | -0.3% |
| Proactive | 1.5% | 1.9% | 1.9% | 1.6% | 1.5% | 1.2% |
| Prospective | 45.1% | 56.0% | 69.6% | 72.1% | 78.3% | 68.1% |
| EWS | 1.3% | 1.6% | 1.6% | 1.4% | 0.5% | 0.7% |
| transfer | 0.6% | 0.3% | 0.1% | -0.2% | -0.4% | -0.4% |
| Credit2 | 3.7% | 3.7% | 2.8% | 2.4% | 1.2% | 1.7% |
| training | 2.8% | 3.3% | 2.2% | 1.7% | 0.9% | 1.3% |

*% change tov 1343 adopted measures under NC reactive*

| Total | NC | Wet | Wet Hot | Hot | DRY | Dry Hot |
|---|---|---|---|---|---|---|
| Reactive | 0% | -1.8% | -5.0% | -8.2% | -18.9% | -15.0% |
| Proactive | 40.0% | 41.2% | 48.2% | 47.6% | 48.8% | 44.8% |
| Prospective | 139.2% | 149.6% | 167.9% | 170.5% | 176.9% | 166 2% |
| EWS | 21.7% | 24.2% | 22.6% | 16.4% | -3.4% | 2.5% |
| transfer | 25.1% | 16.1% | 20.7% | 25.9% | 25.2% | 19.8% |
| Credit2 | 30.2% | 27.3% | 22.3% | 17.7% | 7.9% | 12.9% |
| training | 27.0% | 24.9% | 09.7% | 14.8% | 1.6% | 6.2% |

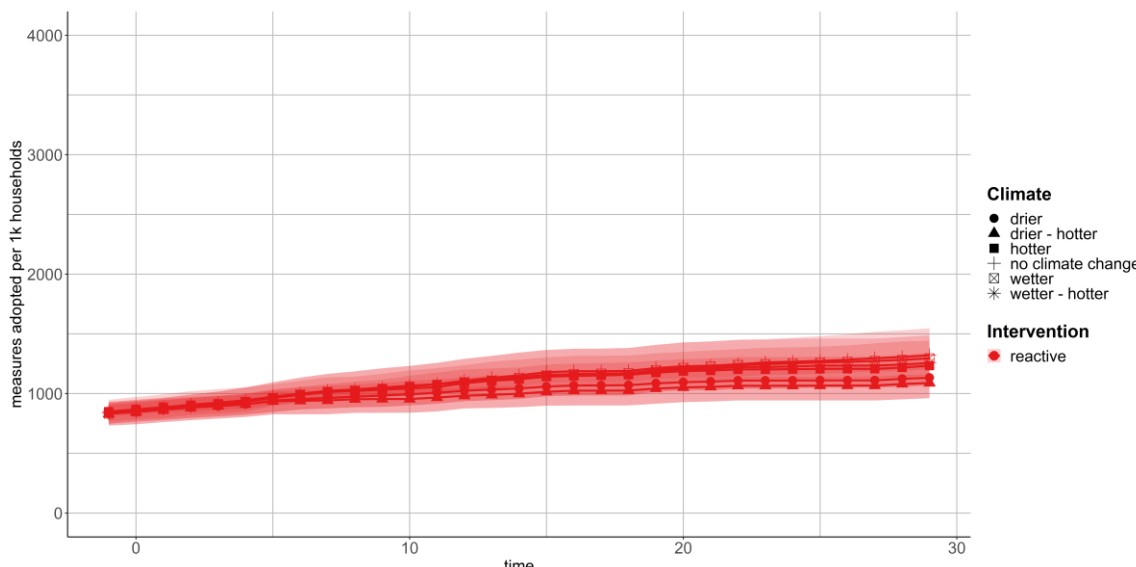

**Figure B1: Total amount of measures adopted per 1000 initialized households under the reactive scenario, averaged over all runs. The shaded area indicates the uncertainty introduced by different model initialisations and by different relative importance of the PMT factors on the decisions of households. Year 0 initiates policy drought risk reduction interventions (indicated with different line colours).**

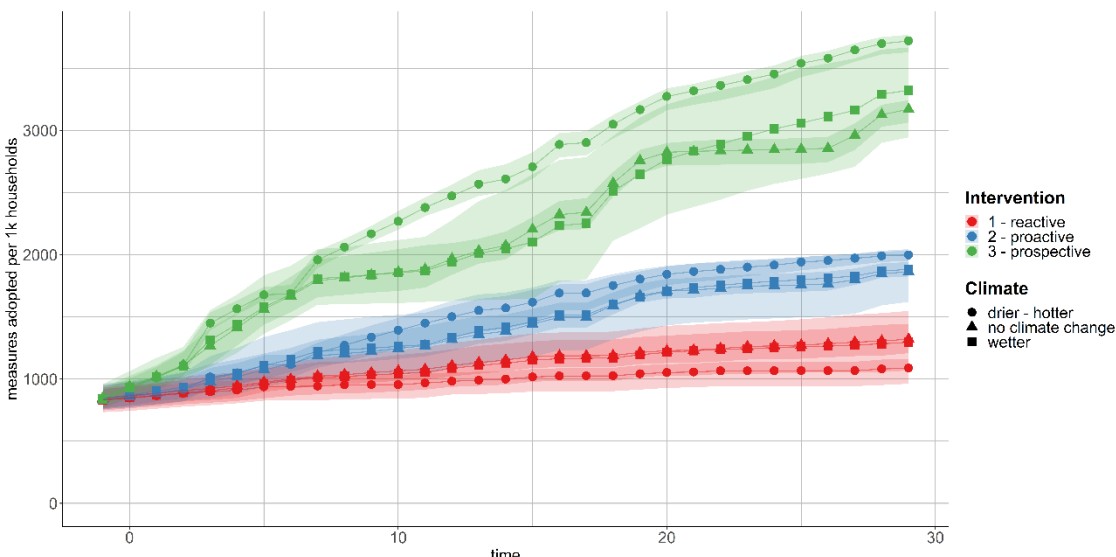

**Figure B2: Total amount of measures adopted per 1000 initialized households under the three intervention scenarios and three climate change scenarios, averaged over all runs. The shaded area indicates the uncertainty introduced by different model initialisations and by different relative importance of the PMT factors on the decisions of households. Year 0 initiates policy drought risk reduction interventions (indicated with different line colours).**

**Author contribution**

M. W. took lead in model development, scenario development and writing the manuscript. T.V. assisted model development, A.v.L. assisted with manuscript writing and both contributed to the scenario development. J.A. was at the basis of the creative process of model setup, development and model application and contributed to the manuscript writing.

**Competing interests**

The authors declare that they have no conflict of interest.

**Acknowledgements**

The authors would like to thank Dr Moses N. Mwangi, Prof. Mary Mburu, Mr. Mutinda Munguti and the entire Sasol staff, for their help in creating the model contexts, discussing the initial model setup and assisting with the data collection to calibrate the model. This research is made possible by the Netherlands Organization for Scientific Research VICI research project number 453-583 13-006 and European Research Council grants nos. 884442 and 948601.

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
