# Peer review of "Education, financial aid and awareness can reduce smallholder farmers' vulnerability to drought under climate change"

_Natural Hazards and Earth System Sciences, 2021_

## Community Comment (CC1)

Dear authors,

I have read your manuscript on modeling smallholder farmers' drought adaptation under climate change and human (non-)interventions with great interest. As far as I understand, this manuscript used not only empirical data but also experimental data to build this ADOPT. The authors made great efforts to describe the model and scenarios in detail. However, when considering my own experience and interests, I still want to discuss the potential limitations or questions with you. I will formulate my concerns with a focus on limitations/comments on "Model and scenario description". At the end of my review, I also point to simple text mistakes.

Limitations/comments on "Model and scenario description"

1. Water Management

You mentioned Water Management practices in 3.1, however, I did not see any detailed explanations except for the irrigation method and the source of irrigation water. I would like to suggest to clarify the irrigation water (non-)sharing method – households have common water resources or each household has a separate well, as well as the stored water volumes in the well. If a couple of households share one common well, how do they distribute the water? These two factors are related to the interactions among farmers and the drought adaptation measures. If the model did not consider these two factors, I would like to suggest you add them to the model, then the model might be closer to reality and help you to make more accurate simulations. If the model already took these two factors into account, I would like to suggest you give a more detailed description, thus, the readers can easily know the water situation of the farmers and can better understand the water management practices.

2. Farmers' adaptation intention

In section 3.1, there is an explanation of four PMT factors. As far as I understand, the PMT you used in the model is to test the farmers' adaptive behaviors according to the weight between 0 and 1. However, the manuscript did not provide any equations/framework to explain the factors in detail. In my opinion, for readers, the manuscript should include: 1) the selection criteria of these four TMP factors; 2) the way of translating the factors into a value between 0 and 1 and how the value defines a farmer's intention to adopt; and 3) is there any meaning of a specific value, for instance, if the weight is 0.7, what does it refer to?

3. ODD+D protocol related concerns

Personally, I really appreciate the way how you formulate the ODD+D protocol. As far as I know, however, there are some details missing in the ODD+D protocol. Firstly, I would like to recommend the following articles:

*Grimm, V., Berger, U., Bastiansen, F., Eliassen, S., Ginot, V., Giske, J., Goss-Custard, J., Grand, T., Heinz, S.K., Huse, G., Huth, A., Jepsen, J.U., Jørgensen, C., Mooij, W.M., Müller, B., Pe'er, G., Piou, C.,*

Railsback, S.F., Robbins, A.M., Robbins, M.M., Rossmanith, E., Rüger, N., Strand, E., Souissi, S., Stillman, R.A., Vabø, R., Visser, U., DeAngelis, D.L., 2006. A standard protocol for describing individual-based and agent-based models. Ecol. Modell. 198, 115–126. https://doi.org/10.1016/j.ecolmodel.2006.04.023

Grimm, V., Berger, U., DeAngelis, D.L., Polhill, J.G., Giske, J., Railsback, S.F., 2010. The ODD protocol: A review and first update. Ecol. Modell. 221, 2760–2768. https://doi.org/10.1016/j.ecolmodel.2010.08.019

Grimm, V., Railsback, S.F., Vincenot, C.E., Berger, U., Gallagher, C., Deangelis, D.L., Edmonds, B., Ge, J., Giske, J., Groeneveld, J., Johnston, A.S.A., Milles, A., Nabe-Nielsen, J., Polhill, J.G., Radchuk, V., Rohwäder, M.S., Stillman, R.A., Thiele, J.C., Ayllón, D., 2020. The ODD protocol for describing agent-based and other simulation models: A second update to improve clarity, replication, and structural realism. Jasss 23. https://doi.org/10.18564/jasss.4259

Grimm et al. made great efforts to structure a full ODD protocol in the above articles. Their contribution may be helpful to frame a better ODD+D protocol.

(1) Entities, state variables, and scales
After I read the description, it is still unclear to me how the state variables of the entities are updated by steps, when, and how they update.

(2) Process overview and scheduling
As far as I see, there are only the names of processes mentioned in this section, but it is lacking a workflow chart. The detail of each process and how the processes connect should be included in this chart.

(3) Submodels
If there are submodels, more details of a model should be explained. The Submodels section should connect to the Process Overview and Scheduling section: the name of the submodels, their principles, used algorithms, equations, and etc. should have a clear description.

I really appreciate seeing the crop parameters used in AquaCropOS. I thought if the authors could provide a complete table that contains the lower bound and upper bound of parameters (or an overview of all parameters) used in NetLogo instead of only listing the initial parameters for farmer households would be better.

In order to correct simple text mistakes, lastly, I would like to suggest the authors to check the spelling and punctuation carefully before final submission. I will list some mistakes I have seen (partly):
'limate' to 'climate'

'it' to 'they'

'in limited' to 'is limited'

'explicitly' to 'explicit'

'decision making' to 'decision-making'

'theory' to 'Theory'

'measures' to 'measure'

delete colon after 'such as'

'countries' to 'country's'

'is' to 'are'; 'has' to 'have'

'stragegies' to 'strategies'

'send' to 'sent'

'Annual' to 'annual'

'producer' to 'producers'; 'consumer' to 'consumers'

the first 'there' to 'it'

'scenarios .' to 'scenarios.'; 'occurring ,' to 'occurring,'

'1 ,' to '1.'

'(Muyanga & Jayne, 2006).' to '(Muyanga & Jayne, 2006),'

'member' to 'members'

'year' to 'years'

---

## Author Response (AR1)

**Reviewer #1 Christian Troost**

Dear authors,

with great interest we have read your manuscript on simulating the effects of climate change and policy intervention on the uptake and success of drought adaptation measures in smallholder farming in Kenya. Assessing the effects of knowledge constraints, communication patterns, risk considerations and self-efficacy that determine the actual adaptation decisions of farmers in combination with more basic economic considerations is a timely and valuable contribution. we appreciate the considerable thought and the considerable effort in empirical data collection that you have put into building this model.

As far as we understand, this article is not about introducing the basic model idea and implementation (which has been done in Wens et al. 2020), but also not yet about a pure policy assessment in which the comparison of the consequences of different policy options is thought reliable enough to directly inform policy decisions. Rather, it is a methodological step in between, intended to illustrate how this previously introduced, but not yet well-established model can be used for policy assessment and discuss its potential contributions and limitations. we concur that this is the appropriate framing for the article given the state of the analysis -- this could be formulated somewhat more clearly in the introduction and used to focus the conclusions

Dear reviewer

Thank you for your kind words and interest in the manuscript. Since the model is not fully validated, the presented application can be seen as a proof-of-concept decision support tool. We agree that this should be clearer in the introduction and conclusion and added the following: (Introduction L79): *"While ADOPT should be subject to additional validation steps in order to more accurately and precisely predict future drought risk, we elaborate the potential of this proof-of-concept model by showcasing the trends in drought risk under risk reduction policies and climate change for a case study in semi-arid Kenya."* (Conclusion L498): *"While we present a proof-of-concept rather than predictive model, the results improve the understanding of future agricultural drought disaster risk under socio-economic, policy and climate trends."* We also extended the discussion chapter (adding section *5.4 Uncertainties and limitations in ADOPT* L452-485) to state more clearly the limitations of the model.

1) One of my main concerns with the manuscript is an insufficient description of essential elements of the model that are key to understand and judge the theoretical and empirical foundation of the simulation results. This concerns both the description in the main text as well as the ODD protocol in the appendix. we referred to Wens et al. 2020, which helped to clarify some questions, but key information remained vague there, too. (In any case, since the model is not well-established and in widespread use, so that you can assume familiarity of the reader with the model, the key assumptions, on which the conclusions rely, should be explained in the article itself and not require reading a second article.)

You model the adoption of adaptation measures as a stochastic decision, in which the probability to adopt is influenced by risk perception, self-efficacy and farmer-perceived benefits and cost of the technology. Policy options affect these determining factors in different ways. Your simulation analysis then calculates the resulting consequences of this assumed influences of policy options to quantitatively compare their overall effect for adoption speed, poverty and food security.

In my understanding, the Protection Motivation Theory that you use as a theoretical starting point does provide you with indications on which explaining factors to include in the model, but it does not provide any further arguments for a specific functional form to quantify and map the influence of policy options on explanatory factors and then map the explanatory factors to a probability. This mapping is the essential and central element underlying the model and must be explained in sufficient detail in the main body of the manuscript. Your current manuscript shows a graph and describes it at the level of "this influences that", but to understand the theoretical and empirical foundation, the reader needs to know:

a) the functional form of the mappings (linear, proportional, or more complex with thresholds, nonlinearities)
b) the quantities assumed for the weights and parameters in that mapping (the basic model and the effect of policy options): what affects what by how much?
c) Most importantly, the justification for those choices: with what reasons did you choose the forms, the values for the parameters and weights (or the boundaries you set on them in uncertainty analysis)? How did you come to them: are they ad hoc assumptions? Is there an underlying well-founded theoretical or structural value for them? Have they been calibrated (chosen such that they yield a desired model outcome) and if yes under which auxiliary assumptions, considering which error distribution, which alternative model formulations, was their identifiability actually assessed? In other words, how strong is the evidence for these key parameters?

In addition, the ODD that you present is lacking much detail. The sequencing of processes and the effects of processes on state variables remains unclear. After reading it we still don't have a clear idea of what happens after what in your model and which state variables are affected and updated when by what. (A few more details on this in part II of my assessment.) The submodels section of the ODD is actually not only for external submodels (like AquacropOS in your case) but should describe the details of the individual processes in the model. While the Processes and Scheduling section in the overview part only names the processes and describes their sequence, the submodels section should contain descriptions of equations, algorithms and parameters of each of these processes (see section 3.7 in Grimm et al. 2010). The ODD should enable readers to replicate your model, which your ODD does not yet achieve.

Grimm, V., et al., 2010. The ODD protocol: A review and first update. Ecological Modelling 221, 2760–2768. https://doi.org/10.1016/j.ecolmodel.2010.08.019

To address your general comment 1, regarding the ODD description, we fully revised the ODD+D in supplementary (Given that we designed an agent-based model, we followed the ODD+D protocol of Muller et al 2013 – which is an extension of the ODD protocol of Grimm et al. 2010.) so as to incorporate your suggestions on incorporating details on the weights, parameters, formulas used and their justifications (p23-40). For example, we gave more detail in the submodels section, adding the PMT equations and rationale of the decision model (the model-ification of the Protection Motivation Theory which was explained in Wens et al 2019 will be added to ODD+D II.ii in the section about how agents make decisions ). Since the crop-water model AquacropOS is an existing, open-access model, published by Foster et al.2017, only its basic principles (and the used parameters) are explained while the details on its equations should be sought in the article of Foster et al.

2) Since your manuscript is intended to exemplify the use of your model as a decision support tool for ex ante policy assessment, we miss a more in-depth discussion about the validity of generalizing your model from the current situation to the future scenarios. This can, as you rightly remark, not come in the form of an empirical verification, but must be a form of structural validation. While we assume that the domain for which AquacropOS has been parameterized covers the climatic conditions that you analyse for the future, this is not self-evident for the parameters and weights used in the adaptation decision model. Given their origin (theoretical or calibrated), can all of their values be considered stable over time and across the structural breaks introduced by climate change? This should at least be shortly discussed in the context of the more detailed description we suggested above.

To address your general comment 2, regarding the validity of generalizing the model from current to future scenarios, we would like to remark that indeed, AquacropOS has been parameterized to cover the future climate conditions – other studies have used the model to estimate current and future maize yields (Wang et al. (2015) A review on the research and application of AquacropOS Model;

- Dissa and Yan (2022) Evaluating climate change impact of rainfed maize production yield in Southern Ethiopia;
- Hsiao et al. (2009) AquacropOS – The FAO crop model to simulate yield response to water: III. Parameterization and testing for maize.

- Irmak et al. (2022) multi-model projections of trade-offs between irrigated and rainfed maize yields under changing climate and future emission scenarios;
- Bwambale and Mourad (2022) Modelling the impact of climate change on maize yield in Victoria Nile sub-basin, Uganda;
- Dale et al (2017) Climate model uncertainty in impact assessment for agriculture: a multi-ensemble case study on maize in sub-Saharan Africa).

Regarding the decision making model, the use of a general, existing theory (Protection Motivation Theory (PMT) (Rogers, 1983; Rogers and Prentice-Dunn, 1997)) for the decision module supports us in assuming the processes behind the decision making are universal and the factors of relevance not to change in the near future (Schrieks et al. (2021) Integrating Behavioural Theories in Agent-Based Models for Agricultural Drought Risk Assessments). We agree with your comment in that, in a totally different society, other parameterisations or other theories might better explain the decision behaviour. Therefore, we only evaluated a limited amount of "future" time given the influence of unavoidable black swan effects - and thus we assume the parameters influencing decisions and adaptive behaviour do not change. Withing the model, there is off course a certain transiency in the values of the factors that affect decisions: every year these values are updated, altering the intention to adopt new measures. They also vary between households (which is why we chose or an ABM approach). Since there is a remaining uncertainty regarding the relative weights of the factors influencing decisions, we did a sensitivity analysis altering these weights (similar to the one in Wens et al 2019). This analysis shows that our model is robust: the variation introduced by the policy and climate scenarios is larger than the model uncertainty in the model parameterisation. To address your comment within the manuscript, we added the following to the discussion L477: *"Lastly, the model application does assume no shifts in the processes underlying weather and human decision making: both the synthetic future weather situation and the decision making processes are based on past observations. To avoid the effect of systemic changes and black swan effect, only 30 "future" years are modelled. "*

3) I am uncertain about some basic economic relationships in your model:

a)  What about the market dynamics in your model. Does your model contain market dynamics at all? Do prices vary at all -- whether exogenously forced or endogenously determined? You speak of the influence of regional supply on household food security, but this seems to be purely quantity-based, without any effect on prices. Supply from outside the region does not seem to be considered. Prices that react to supply shortages would improve income opportunities for crop sellers and complicate matters for crop buyers, in good years that mechanism would work vice versa. Strictly, then also poverty thresholds that are based on valuations of minimal consumption baskets (if they are?) would have to be updated. If not, this should be explained and discussed as a potential limitation.

b) I am unsure how the actual cost and benefits of the adaptation measures enter your model. we can see that they enter as part of the calculation of the probability to adopt and if we understand correctly from your previous article, they can be prohibitive, i.e., lead to probabilities of zero if cost are not covered. Does this only refer to cash demands? Or is there also a consideration of labour availability. How are opportunity costs for labour used on farm instead of off-farm and vice versa? You write that most adaptation measures provide advantages in most years. Does "advantages" here only refer to positive effects on yield, or to an overall positive economic balance after subtracting additional costs from increased revenue? Would such near-certain benefits trigger uptake of measures in your model irrespective of risk appraisal for droughts, or would this every-year benefits be ignored and only the risk mitigation aspect is considered?

c) As far as we understand , in your model, agents always have "receiving food aid" as a backup: Do agents consider this in their risk appraisal? Would decisions change if this food aid was not available (reality and model)?

To address your general comment 3, regarding economic relationships in the model, we agree that this was not fully clear in the manuscript nor ODD+D description and propose to clarify this in the manuscript. We would like to highlight that in our analysis, indeed, only an internal market is considered, with fluctuations responding to local changes in demand and supply. In the revised ODD+D, the following is written L569: *"The market is influenced by local production and consumption, which results in a variable maize price depending on the balance between supply and demand. In the presented case study, we consider relatively isolated areas, less subjected to*

*globalized market systems: maize price is variable following the total amount of locally produced maize to replicate the observed price volatility (with minimum and maximum prices derived from FEWSnet) during years of reduced production."*

No restrictions on labour availability are given, but households with a larger family size have a higher self-efficacy (as children in this region often help on the land). Labour costs in terms of time are unfortunately not included as no clear conclusions could be drawn from the questionnaire on this topic. The price of the measures is derived from their (perceived) costs based on this questionnaire and is included in the adaptation-cost factor. (Both self-efficacy and adaptation cost are part of the coping appraisal factor of the PMT). With respect to the "advantages" we indeed refer to positive effects on yield – but given the protection motivation theory, a certain amount of risk appraisal should be there in order to consider the measures, even if they are cost-effective on the short term and could thus be seen as a general agricultural investment. However, most of them are cost-effective because of the existence of droughts (if we exclude drought years and look only at their impact on crop yield in normal years, they would not be cost effective).

"Receiving food aid" as backup is not considered in the agents' risk appraisal (in the model based on their memory of past shocks) but on the field, from the interviews it was clear that many find the current aid insufficient so we feel it is ok to assume this also does not greatly affect their risk perception (although more research on this could be very interesting). In the model, this aid does not increase households' wealth (it rather avoids households to have negative wealth or to "die" altogether).

4) we am pleased to see that you do an uncertainty analysis to check for the robustness of results. Given the uncertainty in many parameters that you admit, this is essential and communicating the robustness of your findings against this uncertainty strengthen your results. However, your current uncertainty analysis is only vaguely described and remains underexplored.

I can see three different sources of uncertainty that warrant analysis:

   a) The epistemic uncertainty in the weights and parameters in your model
   b) The initialization of the starting population
   c) The sequence of random numbers drawn (aleatory uncertainty)

Please better describe what exactly was varied following which kind of sampling scheme? (I suggest a separate subsection on the design of the uncertainty analysis in the methodology section, currently some incomplete information is scattered over several parts). With respect to the random sequence: How did you control for randomness to ensure that observed difference between scenarios are due to structural differences between the scenarios and not through differences in random draws between runs? (Did you assess a sufficient number of agent decisions so that you can assume convergence/averaging out of random effects or did you employ a common random numbers scheme (see Troost & Berger 2016 for a discussion)?)

In more general terms, while you show uncertainty distribution in a few cases, you mostly communicate single numbers often without indicating whether these constitute averages over repetitions or not. Please always report ranges … and also assess whether the differences between scenarios or ranking of the scenarios remains stable over the repetitions in the uncertainty analysis, or whether their ranking/direction is reversed in some cases. Where uncertainty ranges don't overlap, the robustness of the ranking is clear, but where they do overlap this is not self-evident (see discussion in Berger & Troost 2014, which you already cite, on focusing on the distribution of the pointwise scenario differences rather than the differences between the scenario averages over the distributions).

Troost, C., Berger, T., 2016. Advances in probabilistic and parallel agent-based simulation: Modelling climate change adaptation in agriculture, in: Sauvage, S., Sánchez Pérez, J.-M., Rizzoli, A.E. (Eds.), Proceedings of the 8th International Congress on Environmental Modelling and Software, July 10-14, Toulouse, France.

To address your general comment 4, regarding uncertainty, we agree that it is needed to better describe what was varied through the model runs. We would like to note that indeed, we did account for all three: each run is starts with a random initialisation of the starting synthetic population, and also the random numbers drawn each timestep for each household, is not controlled (they are drawn based on the random-float and random-normal algorithms of Netlogo). The variations in the weights are systematic to allow for a maximum spread. Both are described in Wens et al 2020 and repeated here (L140) *"Through a sensitivity analysis, both the average effect of individual adaptation decisions and its endogenous model variability are analysed (similar to Wens et al 2020).We used, using 12 different initialisations per scenario to include variations in model initialisation, the stochasticity that determines the individual adaptation decisions, and the relative weights of factors influencing behaviour"*. Moreover, since we used an agent set of 1000 households (based on the characteristics of a survey among 260 households); different draws did not have large differences. The sensitivity to the assumptions about adaptation behaviour includes some variability, which is (now) clearly represented in Fig. 5, Fig. 7 and Fig.6 (and in the text of fig.8).We now also explain this more clearly in the ODD+D (II.ii Individual Decision Making, L867).

5) I am somewhat concerned about the sampling you chose to initialize the agent population. You say that you sample based on average and standard deviation, but: Are mean and sd sufficient descriptors of the empirical statistical distribution? Are household assets, family and farm sizes really normally distributed? And are they uncorrelated (looks like you sample individual assets independently, but this is not clear from your statement)? In my experience distributions of farm characteristics are often skewed and correlated. Empirically accurate statistical properties of the starting population are potentially very important in your case, because according to your model description the speed of diffusion will depend on the composition of your model population (e.g., few or many innovating households).

To address your general comment 5, regarding the agent sampling, we again agree this was not well described and detail this in a revision of the ODD+D part on Initialization (III.ii): *"Correlations are present, but the distributions are assumed normally distributed (given the not so large sample of 260 households) except the initial household assets which are based on IFPRI data (table S1)."* If we wanted more empirical households, we could have used the dataset of 260 households to directly initialise 260 agents (which is often done in literature), but this method risks overfitting. We added this as suggestion to the discussion L472:*"Another improvement to the model could be to, instead of initializing the model from distribution functions based on frequencies in the empirical data, directly sample this empirical household survey data (Wens et al 2020) for the creation of a set of agents (households) with different characteristics. Such one-to-one data-driven approach is similar to microsimulation and gaining popularity among ABMs (Hassan et al 2010)."*

To address your specific comments, we changed the following in a revised manuscript:

l. 42f.: "Uncertainties in adaptive behaviour are often addressed by using different adaptation scenarios, but this approach fails to capture the two-way interaction between risk dynamics and adaptive behaviour dynamics (Elshafei, 2016)." This statement is very condensed, not entirely clear what you mean here. Do you contrast fixed exogenous adaptation behaviour with endogenously modelled adaptation? Or do you contrast a few scenarios vs a full global uncertainty analysis?

We now specify this in the text L46: *"Uncertainties in adaptive behaviour are often addressed by using different fixed, exogeneous adaptation scenarios, but…"*

l. 44: we have problems with the use of the term "economically rational" as a contrast to boundedly rational. In my view, it is a misconception that bounded rationality does not constitute economically rational behaviour. we believe this comes from a conflation of the "rational expectations hypothesis", often equated with the assumption of perfect foresight, and the more basic assumption of rational behaviour. The former refers to how expectations are formed (and how expectations relate to true outcomes), the latter how decisions are taken after an appraisal of the situation including forming expectations. It is true that especially aggregated economic models typically have to rely on both assumptions at the same time to analytically derive aggregate equations.* Nevertheless, these two aspects should be kept apart to avoid confusion - especially, since the term 'rational' bears a connotation of "doing the objectively right/the sensible" and the opposite of rational would be

"irrational". Boundedly rational behaviour can well be economically rational given the state of knowledge, expectations and cognitive resources of the agent (Day 2008). we suggest using something like "perfectly optimizing under perfect foresight" or "perfect global optimization" instead. (*It is also true that for a time some orthodox economists may have considered only perfectly informed optimization as proper economic rationality, but there has always been a more fundamental, wider conception of rationality in economics, with the narrower understanding being a subset employed for specific conditions (see e.g., the discussion in Caldwell 1991).)

Caldwell, B.J., 1991. Clarifying Popper. Journal of Economic Literature 29, 1–33.

Day, R., 2008. Micro foundations for meso and macro-economic theory. Journal of Evolutionary Economics 18, 261–273. https://doi.org/10.1007/s00191-007-0084-2

We see bounded rational as context-dependent intuition and reasoning bounded by beliefs, perceptions, and emotion (similar to e.g. (Kahneman & Tversky, 1979)), and therefore would like to stick to this term

l. 52: "innovative" -> In principle, if you claim novelty, you should prove or at least precisely explain it: What had been done before? What is new in your approach? Contrast it to other approaches of modelling farmer decisions ( E.g., Coupling of FAO crop water model (Not AquacropOS but predecessor FAO56) with a boundedly rational ABM and innovation diffusion was done  early in the evolution of¶¶ABMs in Berger (2001) already. There is an overview in Huber et al. 2018, though targeted more at European agriculture.)  But, well, as said above, we guess the novelty of the model should not be the main focus here, but rather its application to climate adaptation analysis and the specific results of the analysis.

Berger, T., 2001. Agent-based Spatial Models Applied to Agriculture: A simulation tool for technology diffusion, resource use changes and policy analysis. Agricultural Economics 25, 245–260.¶¶Huber, R., et al., 2018. Representation of decision-making in European agricultural agent-based models. Agricultural Systems 167, 143–160

We agree that novelty should not be the main focus of the paper so we removed the use of "innovative" in the manuscript.

l. 58: Your wording implies "intention = motivation". Can that be equated?

We agree this is a confusing wording and would change this to L63: *"The PMT suggests that the intention to protect (in this study, the farmers' intention to adopt a new adaptation measure) is motivated by a persons' risk appraisal (their risk perception, experiences of risk events in their social networks…) and the perceived options to cope with risks."*

l. 70: "such as" is a bit vague. What does that mean? Has it been applied exactly to these communities as representatives of this context or to a stylized community that represents communities such as these (then why only mention them and not all for which the stylized community is representative)?

We agree this is a bit vague and removed it. It is now phrased as L84 *The ADOPT model has been applied to the context of the dryland communities in the areas Kitui, Makueni and Machakos in south-eastern Kenya.*

l. 82: Not clear what you want to show with the statement about what farmers "produce ... to ensure adequate supplies": the production potential (this is what they can produce per farm. How large are farms?) or the consumption demand (this is what they need and hence produce. How large are households? How does this combine with bought maize mentioned below?) or the actual production (they could maybe produce more, but they actually produce this. Why?)

L82: We agree this was not clear and would rephrase it to make clear that here were refer to the subsistence farming in the region. It is now rephrased to L97: *Maize is grown in the two rainy seasons, with the aim to meet household food needs (subsistence farming)*

l. 104: what does "calibrated" refer to here? ADOPT as a whole or the AquacropOS part of ADOPT. Clarify. Also, much more details on calibration are needed (for what set of conditions, how transferable/stationary are these conditions compared with the situation analysed in this article, see general comments 1 and 2).

We agree this should be clarified as these concerns indeed AquacropOS – now we refer to the ODD+D where this calibration of AquacropOS is repeated.

l. 106:" which was derived as most suitable in an earlier study (Wens et al, 2020) and has proved to best describe the observed behaviour of farm households (Wens et al 2021)."¶¶-> You should be more cautious in your statements here. In Wens et al, (2020) your specific operationalization of PMT in an heterogeneous ABM formulation was tested only against constant behaviour and expected utility theory (assuming perfect foresight and homogeneous, simplistic preferences) in an non-heterogeneous agent setting, and you concluded "The inclusion of PMT behaviour is thus better able to capture some of the variability in adoption decisions, but is nevertheless still not a complete explanation of the observed adaptive behaviour of households in semi-arid Kenya". From Wens et al. 2021 we don't read a clear preference for PMT over ToPB, CONSUMAT from conclusions and the discussion.¶¶Besides methodological concerns that we would raise towards the deduction of strong conclusions on "best" models based on the methodological setups of these cited studies (confounding of effect of PMT and effects of heterogeneity vs homogeneous agents, lack of assessment of identifiability of contrasted models, lack of explicit error model/likelihood function for empirical tests in the 2020 article, lack of rigorous ex ante definition of which kind of observation would be able to clearly discriminate between contrasted theories) your wording evokes a notion of certainty and comprehensiveness in empirical support for PMT that is not consistent with the earlier manuscript themselves. we would be fine with "better than EUT with perfect foresight", "provided strong arguments for the use of PMT", but not "proved to best describe".

L106: We thank you for remarking this misleading phrasing and would change it to L125:*" The adaptive behaviour of the farm households (agents) is modelled based on the Protection Motivation theory (PMT, Rogers 1975), which was derived as promising in an earlier study (Wens et al, 2020) and includes multiple relevant factors that drive the observed behaviour of farm households (Wens et al 2021)".*

 l. 109: "12 different initialisations to allow for uncertainty in the relative importance of the behavioural factors." <-- this requires much more detail about the uncertainty analysis (see discussion in the General comment 4)

L109: We agree this requires more detail and propose to elaborate this in chapter 3.1 L140: *"Through a sensitivity analysis, both the average effect of individual adaptation decisions and its endogenous model variability are analysed (similar to Wens et al 2020).We used, using 12 different initialisations per scenario to include variations in model initialisation, the stochasticity that determines the individual adaptation decisions, and the relative weights of factors influencing behaviour"* and in the ODD+D III.ii

l. 130f.: How is farm income calculated? Especially, at what prices do you value consumption of self-produced food? ¶¶(Judging from this line in Wens et al. 2020: "any additional harvest is sold (farm income, increasing the households' financial assets)" you seem to actually refer to sales revenue, not precisely income in accounting or economic terms). Is the poverty line fixed in nominal monetary terms or is it based on a consumption basket and recalculated when food prices change?¶¶Do food prices react to supply shortages? Is supply from world markets considered? we could not find any description of an actual market module or food price variation in the model description.

Farm income is seen indeed in a limited way as only income from excess harvest sold on the market. The poverty line is fixed; but food prices react to supply shortages. This allows comparing past and future poverty in a more simplistic way. We added these clarifications regarding food price variation to the ODD+D I.iii.

l. 138: Why are these "proxies" for drought risk? Aren't these the relevant indicators themselves?  Is drought risk an (unmeasurable) independent quantity or isn't it rather a statistical property of such indicators (e.g., the drought risk for income etc.). Or did you mean "proxies" for farmer well-being?

We see drought risk is a complex phenomenon that includes multiple direct and indirect impacts. While the three factors can be seen as indicators, we feel they rather approximate some aspects of risk, hence the word choice.

l. 151f / l. 160ff: Too vague: What does added mean? Was a 365th part added to each day of the year? Or was it unevenly distributed over the days of the year? You are already defensive about the scenarios, but since you focus on droughts as extreme events the former would be a very coarse, potentially biased allocation of changes. Not sure how relevant it is as it depends on the sensitivity of the AquacropOS model to such imprecision. You should explicitly discuss the potential implications/robustness of such a simple addition on AquacropOS results.

We agree this is too vague. It is indeed the former (the daily values of the past are altered by adding / subtracting the 10%/15%), and we know it is quite biased (as this ignores the increase in variability) - it is a very simplistic way to create synthetic data). However, we wanted to keep the day-to-day and dekad-to-dekad correlation and variability as this is important for AquacropOS while avoiding gridded data that has includes an averaging effect (important for rainfed agriculture). We do not claim that these scenarios have a certain known probability of occurring; it is rather a "what if the same happened under warmer / wetter /… conditions". We will rephrase it as follows L181: *" These trends were added to time series of 30 years of observed data. While this approach does not account for an increased variability, it allows to account for the temporal coherence in the data (e.g. persistence) and the interrelationships among different weather variables (weather generators – another option to downscale projected climate - have still some progress to make in order to accurately account for extreme events (Ailliot et al., 2015; Mehan et al., 2017).."*

¶¶Fig. 5: The graph should be improved. While one can clearly distinguish the strategic and proactive scenarios from the other scenarios, the remaining scenarios are so close together that one can see that the climate scenario seems to make a difference, but the details and directions cannot really be discerned. It would also be much more understandable if the shorthands for the interventions used in the graph had been mentioned in the earlier scenario description as used here. Please not in the caption, how the values have been aggregated over the uncertainty runs.

You show the number of adopted measures, but since it is higher than the number of households, many households seem to adopt several measures. Just the number of measures may however not be a clear indicator of the strength or effectiveness of adoption. Three low-cost, low barrier, low protection measures may actually be indicative of less motivation and effectiveness in adaptation than the adoption of one high cost, high barrier, high effect measure. You should reflect and discuss this somehow or justify why the number of measures is a good indicator in your case.

We improved the graph (fig5) by using the full name of the interventions and reducing the number of lines (two complementing figures are in the Appendix B to cover all information) to show the variability. We agree the sum of measures does not tell all. Nor would the number of households with at least one measure, or the number of households with the most effective setup – rather it is a combination of these that is interesting. Since it is impossible to visualise all, this graph only shows the sum, but the table in appendix B gives the breakdown per type of adaptation measures. Data on the amount of households with x amount of measures exists as well, but we found that even harder to interpret / draw clear conclusions from so decided not to add this to the manuscript.

l. 212: "This means that adaptation intention is indeed limited by a low risk¶¶perception, high (initial) adaptation costs, a limited knowledge of the adaptation efficacy or a low self-efficacy." -> That is already an interpretation and should probably better go to the discussion section, rather than the result presentation. Also, it remains very coarse. It would be interesting to get some more information on which of these factors that you enumerate is most important.

L212: We would argue it is not really an interpretation as this is really the core of what is in the model (and one cannot draw conclusions about something that is internalised in the model itself. However, we phrased it a bit better hoping it reads more like results now L257: *In the reactive scenario, adaptation intention is clearly limited by a low risk perception, high (initial) adaptation costs, a limited knowledge of the adaptation efficacy or a low self-efficacy. Some of these barriers are alleviated through the different government interventions*. In the discussion (5.1), we repeat this observation while adding a more detailed interpretation.

l.223: Instead of the clumsy "(non-)governmental", why not leave this out (just "intervention") or use "policy intervention"? Since both non-governmental and governmental are meant, basically "all" outside interventions

are meant and there is no need to constantly (non-)distinguish them. Alternatively, the established distinction between "autonomous" (by farmers themselves) and "planned" (by government and non-governmental actors) adaptation could be very helpful here (see Easterling et al. 2007).

We thank you for this suggestion and would use top-down intervention or policy intervention in the revised manuscript.

l. 235: "surprisingly" -> maybe not really surprising given the arguments in this sentence.

We agree with your observation. It was a result that was not expected by the authors, but indeed clearly explainable thus we removed it.

l. 266: "Droughts, climate¶¶change and adaptation levels influence the availability of maize on this market."-> Is there a market mechanism in the model? Is there endogenous market supply and closure?

Yes, there is a limited form of market working (see previous repones regarding this topic)

l. 356: "Because results of the future scenario runs cannot be falsified or verified, this study claims not to provide a prediction of the future for south-eastern Kenya". Results for the future can never be verified before the future becomes the present (and often not even then, due to limited control of outside influence in open systems). That's the nature of prediction. In this sense, the reason you give here misses the point. we fully agree that your simulations do not provide forecasts, i.e., point predictions of the future. But the reason is that the system you are predicting is, on the one hand, unstable and not fully understood, and on the other hand, too open to outside influence that cannot be controlled in a few exogenous scenarios (which is underlined by the uncertainty in those scenarios you already have). So, the system is simply not fully predictable. The question of verification and validity concerns the model and simulation setup rather than the scenario runs for the future.

You are absolutely right. We rephrased this to L480: *"Because the model setup could not be fully validated, and scenarios do not provide a complete overview of all possibilities, this study does not claim to provide a prediction of the future for south-eastern Kenya".*

l. 359: "Future research can use ADOPT to study the differentiated effect of these interventions on¶¶different types of households, in order to tailor strategies and target the right beneficiaries of government interventions." This is the correct target for your simulations and should be the focus of discussing the limitations of your study. Mention why you can't provide a point prediction (see above, you already provide essential arguments), but focus the main part of your discussion on limitations towards this actual goal of your modelling (which is not the forecast): What are the limitations of your model with respect to contrasting the effect of different policy options under different circumstances? What are the limitations in highlighting and differentiating vulnerability and heterogeneity between farmers?

Thank you for this suggestion. In the discussion, we extended the section on uncertainties and limitations (now L452-485)  to include information of what the model does and what it cannot do (and why).

l. 376: "We show that all investigated interventions have a positive effect on the uptake of adaptation measures," Was that in doubt? Or better: would your model -- by design -- allow for one of these interventions to slow down uptake? (Please discuss this in the discussion section and, if it does not, don't emphasize this as a major outcome, but directly come to the quantification of the effects. )

It could be that some interventions alleviated a barrier that was not the main barrier and thus show no effect – which would mismatch the conclusions of (Wens et al. 2021). We now elaborate this in the (rewritten) discussion section part 5.1 L367.

l. 377ff: "Extension¶¶services increase the adoption of low-cost, unknown drought adaptation measures while credit schemes are useful for cost-¶¶effective but expensive drought adaptation measures. Ex-ante cash transfers allow the least endowed households to adopt low-¶¶cost popular drought adaptation measures. Early warning systems are more effective in climate scenarios with less frequent¶¶drought if used as a tool to create

awareness and risk perception." These statements are very general, and most people would subscribe to them before having seen your model or its outcomes. Could you conclude more precisely? What exactly could be seen in your model results that is not self-evident from the assumptions that you put in the model?

We think the relative effect of the measures because of their diversified effect on the heterogeneous households is an interesting confirmation of what we would indeed expect. We now describe this more precisely by adding numbers to this part of the text (chapter 5.1).

l. 386: estimates → predicts or implies

We agree with your suggested change

l. 388: "this study proves" → Prove is too strong: It would require proving that all the assumptions of your model are fully correct and a proof of a complete incorporation of all relevant systematic effects in your model, which you admit is not possible. The results "suggest" or "shows that under the following assumptions..."

We agree with your suggested change

With respect to your comments on the ODD+D in appendix, we would like to propose to greatly revise this section to account for all your comments and suggestions.

ODD I.i "The ADOPT model...management policies" This is a description of what the model is supposedly capable of, but not what the actual purpose is. To really judge whether it fulfils the intended purpose, this purpose must be formulated much more precisely:¶¶ To me it seems that the purpose is¶¶  - to simulate the (welfare, income, ..?) of smallholder farming households as a function of climate effects on agricultural production mitigated adaptation decisions.¶¶  - to simulate the speed of uptake of adaptation measures by these households as a function of policy interventions, household characteristics, ...¶¶  --> in this way, to allow identifying vulnerability by detecting where adaptation does not suffice and contrasting the effects of different policy options

Thank you for your suggestion. We included this in the rewritten ODD

ODD I.ii: ¶¶- "stock of assets": could you name them (at least with examples)?

- More precisely: we would say that "weather" is an exogenous factor, not climate change. Climate change is reflected through different input scenarios for this external factor weather. (Or do you model weather endogenously?)\

- What about prices? Are they exogenous (then they should be mentioned here) or are they endogenous? (Then is there an auctioneer agent or market agent or and entities/states that represent the market that should be mentioned in entities?)

We phrase this and remove the use of "stock of assets". We indeed see weather as an exogenous factor while climate change is reflected through different scenarios altering this external weather factor. Prices on the other hand are endogenous, based on supply and demand within the model. We created an UML diagram in the appendix (ODD+D) to clarify this.

ODD I.iii: This is a bit too short and coarse. Please also try to formulate this in terms of the state variables and entities that you mentioned in I.ii and as actual process steps with timing/sequencing (this does that then, then this happens) - not as general relations with unspecified timing (this influences this ) (E.g., is there a cash reserve (state)? How is it changed at what point of time?)

 1. The agent plans based on ...¶¶ 2. The agent implements ... ¶¶ 3. Crop yield is determined ...¶¶ 4. Agent harvests, uses income... cash reserve is updated, assets ..., ¶¶ 5. poverty or food security determined? ¶¶ 6. Agent updates expectations (interact with neighbours? Who interacts first? Or no sequence but pools of

interactions?)¶¶ ¶¶ or something like this ( A bit in the style of the "ADOPT runs as follow" enumeration in Wens et al. 2020 but extended to the full modelling sequence.)¶¶

We extended this section and added a flowchart to support the description

II.ii: "Do agents pursue an¶¶explicit objective or have other success criteria? How do agents make their decisions?" These questions are not yet answered in your text. Do agents have a goal that is explicitly coded into the model and select among options with respect to a success criterion (objective function) reflecting the goals (e.g., optimization, satisficing, prioritisation)? Or do they act following an empirical function/probability that states a probability of a certain action under certain circumstances? (Without an explicit modelling of goal considerations/success criteria in the model.)

We extensively altered this section to include  more details on how decisions are coded in the model

II.viii: "Okumu (2013),¶¶Shikuku (2017) – among others - found that state variables such as age, gender,¶¶education of the household head and the household size have significant effects on this¶¶risk-attitude" -> The question is not what others found, but what you implemented in the model. Which of these variables influences decisions in the model and how?

We rephrased this to focus on the model implementation

III.ii: "What is the initial state of the model world, i.e., at time t=0 of a simulation run? At the¶¶initial stage, households and their characteristics are randomly created based on the¶¶mean and standard deviation derived from the household dataset" -> See general comment number 5. In addition, clarify whether  characteristics of a household are sampled independently from each other or whether correlations are preserved.

We provided means and sds of these initial variables in Table A1 and A2 and added information on dependencies in the text.

III.iii: The Input Data section of the ODD always creates confusion, but it actually refers to time series of exogenous variables that drive the model and influence the system over time (not to all input data used somewhere in the model). (See Grimm et al. 2010)

Thank you for this clarification, we changed this section accordingly

III.iv: This is way too short. The submodels part should give the details for each of the processes mentioned in I.iii. So ¶¶if in I.iii there is a process "Agent decides on uptake of measure ...", then here a corresponding subheading with the mechanisms and key equations used to model the decision should be included  (see Grimm et al. 2010).

We feel it is not needed to explain an existing, open-access standalone model that is published (AquacropOS) so we just refer to the academic literature regarding this model. The other 'submodel' is the decision module but this one is already explained in quite detail in the part above (it is not really a submodel, rather the core of ADOPT) so we feel also here, not much additional info would be needed.

¶¶III ORTHOGRAPHY AND LANGUAGE

l. 27: "limate" -> Climate¶¶

l. 29: depends -> depend (plural)¶¶

l. 30: "re-occurring" -> recurring or repeated¶¶

l. 44: bounded -> boundedly¶¶

l. 58: comma missing after "in this study", also "the" should be added before "farmers'"¶¶

l. 59: remove comma here: "), is"¶¶

l. 60+61: comma missing before "...". But better leave out the "..."¶¶

l. 76: countries' -> country's (genitive singular)¶¶

l. 78: omit "for the people"¶¶

l. 80: "smallholder farm maize crop yields" -> omit farm and crop -> "smallholder maize yields"¶¶

l. 80: insert comma between management and maize yield¶¶

l. 80: maize yields¶¶

l. 82: please don't jump between units, translate to tons or kg.¶¶

l. 122: thrust -> trust¶¶

l. 130: Annual -> annual¶¶

l. 175: comma missing after parenthesis¶¶

l. 177: remove comma, after parenthesis¶¶

l.224: an strong -> a strong¶¶

l.230: "had the is highest effect" -> remove "is"¶¶

l.231: "droughts measures" -> drought measures or better drought adaptation measures¶¶

l.254: with respect to all the % mentioned: is it by 5% or by 5 percentage points?¶ Percentage (change in yield from past to projected scenario) in this section, but indeed percentage points in the following sections.

l.256: "with 11%" -> "by" or "to"? ¶¶ by

l. 304: wrong placement of parenthesis. Either instead of comma before as, or only around the years.¶¶

l. 342: missing space between "al." and "2021"¶¶

l. 516,520,540, and other books and you cite: These references miss publisher and location of publication.¶¶

l. 527: "Berger, T., Wossen, T., Troost, C., Latynskiy, E., Tesfaye, K., & Gbegbelegbe, S. (2015)..." ¶¶-> This conference paper was superseded by a full journal publication, which you might prefer to cite instead (Berger et al. 2017, Can smallholder farmers adapt to climate variability, and how effective are policy interventions? Agent-based simulation results for Ethiopia. Agricultural Economics 48, 693–706. https://doi.org/10.1111/agec.12367)¶¶

l. 746: The Doi link provided (https://doi.org/10.1002/eqe.3063) does not point to the indicated reference, maybe this is the correct one: https://doi.org/10.1002/wat2.1345 ?

We greatly appreciate your extensive comments, which we addressed above, and are sure implementing them will improve the manuscript significantly. We would also like to thank you for your detailed orthographic and language review, which we will consider in a revised manuscript.

Respectfully,
Marthe Wens

Reviewer #2 Anonymous
The paper deals with an interesting topic and is the results of a large research effort by the authors. However, the presentation of the authors' work needs to be improved in several aspects. The study is quite complex and for this reason special care and effort are needed to make it understandable to the reader. we have some major/general comments and then other comments that refer to specific sentences in the text.

Dear reviewer,

Thank you for acknowledging the complexity of the topic of and your interest in this manuscript. We greatly appreciate your detailed feedback.

A better explanation is needed of where this research stands in the literature on the topic and particularly in the body of research by the same authors. E.g., what does this paper add in relation to Wens et al 2020 and Wens et al 2021? Is the contribution of this paper methodological or to the definition of policies in a specific place? Is the ADOPT model something developed by the authors and for this paper? Once this is clarified, the paper should be revised accordingly, in order to make that contribution emerge more clearly throughout the text.

To address your first general comment, we clarify the position of this research in the introduction as follows L71: *"In this study, we apply ADOPT, to test the variation in household drought risk under different drought management policies: (i) a reactive government only providing emergency aid, (ii) a pro-active government, which supports ex-ante cash transfer in the face of droughts and sufficient drought early warnings, and (ii) a prospective government that supports adaptation credit schemes and provides regular drought adaptation extension services to farmers. In addition, ADOPT is used to evaluate drought risk and the robustness of these policies under different climate change scenarios. The design of ADOPT as an agent-based drought risk adaptation model is described in Wens et al., 2020, while Wens et al. 2021 detail the empiric data on pasta adaptive behaviour, used to calibrate the model, as well as empiric data on adaptation intentions that can be used to compare with the model outputs. While ADOPT should be subject to additional validation steps in order to more accurately and precisely predict future drought risk, we elaborate the potential of this proof-of-concept model by showcasing the trends in drought risk under risk reduction and climate change for a case study in semi-arid Kenya".*

Section 3: the authors should provide more information about the dataset used (e.g., how was the dataset obtained? who was the surveyed/interviewed population? How was it selected?). The dataset is key to understand the paper and its description cannot be referred to other papers.

To address your second general comment, regarding section 3, we agree that some more information about the dataset is necessary. The data used to parameterize and partially calibrate ADOPT, are indeed described in detail in Wens et al. 2021. We added the most important variables from this dataset to the ODD+D (tables, text) and added the following to the model description in the main manuscript L114: *"ADOPT was parameterized with information from expert interviews, a farm household survey with 260 households including a semi-structured questionnaire and a discrete choice experiment (a quantitative method to elicit preferences from participants without directly asking them to state their preferred options) executed in the Kitui Region, Kenya (Wens et al. 2021)".*

Section 4 Results' structure is confusing. The presentation of the results is very difficult to follow and needs major revision. Maybe the authors could use subheadings and/or tables to guide the reader? Also, some sentences are long and difficult to understand (e.g., lines 216-219; lines 223-225; 231-233). In other cases, it is unclear how the authors came to a given finding/conclusion (e.g. (211-212): some sentences seem to be interpretations of the authors and therefore they should be presented in the discussion section and justified. Figures 5, 6 and 7 seem to transmit the same message (i.e., that strategic plan is more effective than proactive plan and this in turn is more effective than the reactive one) so maybe they are not all needed. The information about climate scenarios is difficult to read in the figures. Also, the authors should check the journal specifications about figures that should be readable both in colour and in B&W.

To address your third general comment, regarding section 4, we agree that the structure is confusing. We added subheadings according to your suggestion to improve the clarity. Besides, multiple longer sentences were identified and improved. We disagree that figures 5,6 and 7 are obsolete. While indeed one message (prospective

intervention has the most promising effect on drought disaster risk reduction) stands out in all three, many other messages (e.g., also climate change does change the adoption rate and more significantly the drought risk, average harvest still decreases under proactive intervention; poverty reduction in the proactive scenario is only distinctly different from the reactive one after multiple years) would be lost if we remove these figures. We improved the figures to align with the journals' specifications and to facilitate reading information about the climate change scenarios.

Section 5. The discussion should be strengthened and made easier to follow as one gets lost in details and long sentences. Are the results presented in section 4 what the authors expected? If not, what is the reason for that according to the authors? If yes, what does the current paper add to the existing knowledge about the topic

To address your fourth general comment, regarding section 5, we added multiple interpretations to the discussion (we significantly rewrote this section and split it in 5.1, 5.2) to improve the content. Besides, we adjusted confusing sentences so as to make this section easier to read. With respect to your second remark: similarities with other studies are pointed out repeatedly (e.g. in 5.1: Hartwich et al., 2008a; van Duinen et al., 2016a; Villanueva et al., 2016; Wossen et al., 2013; Enfors & Gordon, 2008; Mango et al., 2009; Mosberg & Eriksen, 2015; Sherwood, 2013 Gebrehiwot & van der Veen, 2015; Holden, 2015; Makoti & Waswa, 2015; Mude et al., 2007; Oluoko-Odingo, 2011; Winsen et al., 2016; Wamari et al., 2007; Aker, 2011; Wossen & Berger, 2015; (Asfaw et al., 2017; Davis et al., 2016; Pople et al., 2021).

Section 6 Conclusion. The main messages emerging from the application of ADOPT are difficult to identify in this section. As a suggestion, the authors could use the intervention scenarios to guide their conclusion instead of breaking down their messages in long sentences that list different types of drought adaptation measures, barriers, interventions etc. For instance, from the conclusion the reader does not get a key message that emerges from the figures in the results, i.e., that after only 10 years, drought risk reduction is much higher and stable in the strategic interventions scenario than in the other two scenarios.

To address your fifth general comment, regarding section 6, we shortened and merged the first two Alinea's to highlight better the main messages (L487). We added key results that were missing and avoided long complex sentences to improve clarity of this section.

To address your specific comments, we propose the following:

Abstract. What does "(non-) governmental" mean in this context? It should be specified the first time this term is used.

We changed (non-)governmental into "top down" throughout the manuscript

Line 27: please revise the use of "erratic" or "inadequate" adjective to define rainfall. These terms suggest that there is an "adequate" rainfall pattern, which that is a human construct

We agree that inadequate and erratic are human constructs, as are disasters. Because human expectations are not matched, there is a drought disaster. Given this, we think it is best to keep erratic and inadequate.

p. 4 please add a map of Africa to locate the study area in a larger geographical context.

We added a map of Africa

line 96 please spell ODD+D out

We spell out ODD+D on L112

line 99 please explain what a choice experiment is.

We will add the following to explain DCE L116: *"a quantitative method to elicit preferences from participants without directly asking them to state their preferred options"*. More details can be found in the cited Wens et al. 2021

line 107: the expression "has proved to best describe" seems to overstate the goodness of the approach. The PMT may have performed well in the cited research but there is no way to say that "best describe".

We changed this sentence to L59 : *Combining risk models with an agent-based approach is thus a promising way to analyse drought risk, and the evolution of it through time, in a more realistic way (Wens et al., 2019).*

line 165-166 : sentence is unclear

We changed L198 to better reflect fig3: *"Under the no change scenario, 25% of the thirty simulated years fall below this threshold. Under the wet scenario, less such droughts occur (15% of the years), but under the dry scenario, the number of drought years more than doubles (54% of the years). Temperature is dominant over precipitation is determining drought conditions, as under the hot-wet scenario, 41% drought years are recorded, and under hot-dry conditions, 78% of the years can be considered drought years."*

line 314 "a different effect" of what?

We rephrased it to L412 : *"The diverse climate change scenarios have a distinctly different effect on the evolution of drought risk in the rural communities."*

Line 319-321: why do the authors think there is this different in effects in drier climate and hotter climates? What is the assumption in terms of adoption of measures any farmers and implementation of interventions?

We want to note that in AquacropOS, daily biomass growth depends both on temperature and precipitation. Under a hotter and drier climate, the balance between evaporative demand and moisture supply will be different than in scenarios with only hotter or only drier conditions.

Line 329: why the proactive government scenario is not useful under dry conditions? On line 336, the authors say that it reduces emergency aid under all possible scenarios

L329 discusses the proactive scenario, while L336 discusses the strategic scenario – in the reviewed manuscript this will be called "prospective" to match the terminology of GAR 21 SR on Drought.

Lines 178-183: the description of these findings in unclear. What does this paper add to the findings of Wens et al 2021?

Assuming the comment about L178-183 is about L378-383: Wens et al. 2021 shows a data-driven econometric analysis of empiric data, while this manuscript shows a model-implementation based on existing theory. While only the historic behaviour in Wens et al 2021 is used to parameterise the model, the choice model results about potential future adaptation can be used to compare the model outputs (now this is referred to in the discussion). Besides, while Wens et al 2021 could only make hypotheses about the increase in adoption under policy changes, ADOPT model is able to estimate the effect of this on the drought risk of individual households and communities in terms of poverty, aid needs and food security – and this under different climate scenarios.

Section 6 - conclusions: the first two paragraphs are repetitive.

We rewrote the first two paragraphs of the conclusion - merge them into one L487*: To increase the resilience of smallholder farmers to droughts, top-down interventions are needed to alleviate barriers to adaptation, increasing farmers' intention to adopt drought adaptation measures. However, to which extent these interventions will steer farmers' adaptive behaviour, hence how effective they are in reducing the farm household drought risk, often remains unknown. In this study, the agent-based drought risk model ADOPT is applied to evaluate the effect of potential future scenarios regarding climate change and policy interventions on agricultural drought risk in south-eastern Kenya. The smallholder farmers in this region face barriers to adopt drought adaptation measures such as mulching, fanya juu terraces, shallow wells, and drip irrigation, to stabilize production and income.*

Line 354: in several parts of the text the authors talk about "cost-effective interventions": how could they assess cost effectiveness if they didn't know their costs?

We would like to highlight that the cost of the on-farm individual adaptation measures is considered (cost for the farmers themselves, also detailed in Wens et al 2019) – but the cost of the top-down interventions (e.g., extension services) – thus costs for NGOs or government - is not - and clarify this in the ODD+D.

Line 357. we am unsure that it can be considered a decision support tool if it is not predictive.

In our opinion, ADOPT is a decision support tool that provides insight into the effects of measures on drought risks. The predictive element is not important here, but we did look at the trends in impacts under future conditions which is different than trying to foresee actual impacts in actual future years.

Line 378. what does "unknown drought adaptation measures" mean? Unknown to/by whom? "cost-effective but expensive" is an unclear concept: maybe the author means that they are more expensive compared to other measures? Expensive should be in relation to something, it is not an absolute term.

This section (L511 onwards) is rewritten and the wording "unknown" avoided.

We thank you very much for your constructive feedback and are sure that implementing this will greatly improve the manuscript.

Respectfully,

Marthe Wens

Community referee #1 Dengxiao Lang
Dear authors,

I have read your manuscript on modelling smallholder farmers' drought adaptation under climate change and human (non-)interventions with great interest. As far as we understand, this manuscript used not only empirical data but also experimental data to build this ADOPT. The authors made great efforts to describe the model and scenarios in detail. However, when considering my own experience and interests, we still want to discuss the potential limitations or questions with you. we will formulate my concerns with a focus on limitations/comments on "Model and scenario description". At the end of my review, we also point to simple text mistakes.

Dear Dr Lang

Thank you for your interest in and feedback on our manuscript. Based on your review, we rewrote the ODD+D protocol in supplementary to increase the level of detail and clarify many of the model assumptions and design characteristics as per your suggestions.

You mentioned Water Management practices in 3.1, however, we did not see any detailed explanations except for the irrigation method and the source of irrigation water. we would like to suggest clarifying the irrigation water (non-)sharing method – households have common water resources, or each household has a separate well, as well as the stored water volumes in the well. If a couple of households share one common well, how do they distribute the water? These two factors are related to the interactions among farmers and the drought adaptation measures. If the model did not consider these two factors, we would like to suggest you add them to the model, then the model might be closer to reality and help you to make more accurate simulations. If the model already took these two factors into account, we would like to suggest you give a more detailed description, thus, the readers can easily know the water situation of the farmers and can better understand the water management practices.

With respect to water management, we agree this creates confusion and propose to change the term to "drought adaptation measures" as they were used interchangeably in the manuscript. Currently, all considered measures are individual and on-farm applications – no sharing is assumed to happen. we agree looking into community water harvesting structures such as larger boreholes or sand dams would be super interesting, but this requires – as you remark – other interactions on which we did not collect any empirical information. Some lines related to this are also added to the discussion chapter (5.4 limitations on ADOPT).

In section 3.1, there is an explanation of four PMT factors. As far as we understand, the PMT you used in the model is to test the farmers' adaptive behaviours according to the weight between 0 and 1. However, the manuscript did not provide any equations/framework to explain the factors in detail. In my opinion, for readers, the manuscript should include: 1) the selection criteria of these four TMP factors; 2) the way of translating the factors into a value between 0 and 1 and how the value defines a farmer's intention to adopt; and 3) is there any meaning of a specific value, for instance, if the weight is 0.7, what does it refer to?

With respect to farmers' adaptation intention, we would like to highlight that the detailed model design including all formulas is already published in Wens et al 2020 - and the ODD+D attached to this manuscript supplies the most important information regarding the model application of the PMT. We however agree that the formulas should be repeated in the ODD+D of this manuscript; and thus, we significantly revised its description (especially II.ii).

some details missing in the ODD+D protocol. Firstly, we would like to recommend the following articles:

*Grimm, V., Berger, U., Bastiansen, F., Eliassen, S., Ginot, V., Giske, J., Goss-Custard, J., Grand, T., Heinz, S.K., Huse, G., Huth, A., Jepsen, J.U., Jørgensen, C., Mooij, W.M., Müller, B., Pe'er, G., Piou, C., Railsback, S.F., Robbins, A.M., Robbins, M.M., Rossmanith, E., Rüger, N., Strand, E., Souissi, S., Stillman, R.A., Vabø, R., Visser, U., DeAngelis, D.L., 2006. A standard protocol for describing individual-based and agent-based models. Ecol. Modell. 198, 115–126. https://doi.org/10.1016/j.ecolmodel.2006.04.023*

*Grimm, V., Berger, U., DeAngelis, D.L., Polhill, J.G., Giske, J., Railsback, S.F., 2010. The ODD protocol: A review and first update. Ecol. Modell. 221, 2760–2768. https://doi.org/10.1016/j.ecolmodel.2010.08.019*

*Grimm, V., Railsback, S.F., Vincenot, C.E., Berger, U., Gallagher, C., Deangelis, D.L., Edmonds, B., Ge, J., Giske, J., Groeneveld, J., Johnston, A.S.A., Milles, A., Nabe-Nielsen, J., Polhill, J.G., Radchuk, V., Rohwäder, M.S., Stillman, R.A., Thiele, J.C., Ayllón, D.,*

*2020. The ODD protocol for describing agent-based and other simulation models: A second update to improve clarity, replication, and structural realism. Jasss 23. https://doi.org/10.18564/jasss.4259*

Grimm et al. made great efforts to structure a full ODD protocol in the above articles. Their contribution may be helpful to frame a better ODD+D protocol.

(1) Entities, state variables, and scales

After we read the description, it is still unclear to me how the state variables of the entities are updated by steps, when, and how they update.

(2) Process overview and scheduling

As far as we see, there are only the names of processes mentioned in this section, but it is lacking a workflow chart. The detail of each process and how the processes connect should be included in this chart.

(3) Submodels

If there are submodels, more details of a model should be explained. The Submodels section should connect to the Process Overview and Scheduling section: the name of the submodels, their principles, used algorithms, equations, and etc. should have a clear description.

I really appreciate seeing the crop parameters used in AquacropOS. we thought if the authors could provide a complete table that contains the lower bound and upper bound of parameters (or an overview of all parameters) used in NetLogo instead of only listing the initial parameters for farmer households would be better.

With respect to the ODD+D protocol, we would like to note that, given that we designed an agent-based model, we followed the ODD+D protocol of Muller et al 2013 – which is an extension of the ODD protocol of Grimm et al. However, based on your suggestions, we fully revised the ODD+D to reply to increase the level of detail. We address your comment 1 (entities, state variables, and scales) in the process overview (I.iii) by adding a UML diagram; we address comment 2 (process overview) by adding a flowchart; we address comment 3 (submodels) by adding more details about the decision module (in II.ii) but feel that, since the crop-water model AquacropOS is an existing model, published by Foster et al. 2017, only its basic principles are explained while the details should be sought in the article of Foster et al.; and we address comment 4 (initialisation parameters) by adding a table on the agent characteristics in III.II.

In order to correct simple text mistakes, lastly, we would like to suggest the authors to check the spelling and punctuation carefully before final submission. we will list some mistakes we have seen (partly):

**(we improved al of the mistakes)**

27 'limate' to 'climate'

40 'it' to 'they'

44 'in limited' to 'is limited'

48 'explicitly' to 'explicit'

54 'decision making' to 'decision-making'

55 'theory' to 'Theory'

59 'measures' to 'measure'

60 delete colon after 'such as'

75 'countries' to 'country's'

86 'is' to 'are'; 'has' to 'have'

88 'stragegies' to 'strategies'

122 'send' to 'sent'

130 'Annual' to 'annual'

131 'producer' to 'producers'; 'consumer' to 'consumers'

134 the first 'there' to 'it' **(I think here there is ok)**

153 'scenarios .' to 'scenarios.'; 'occurring ,' to 'occurring,' **(unclear suggestion)**

160 '1 ,' to '1.' **(unclear suggestion; 1 instead of "year 0"?)**

175 '(Muyanga & Jayne, 2006).' to '(Muyanga & Jayne, 2006),'

182 'member' to 'members'

196 'year' to 'years'

We would like to thank you for your thorough spelling and punctuation check, and will revise the manuscript accordingly.

Respectfully,

Marthe Wens

---

## Author Response (AR2)

Dear editor

Thank you very much for your feedback on the manuscript. We followed your advice and solved all the issues that you highlighted, and agree the manuscript improved through this round of revision.

- In the introduction, we changed the order of the names of the measures to improve clarity.
  *"The robustness of additional extension services, lowered credit rates, ex-ante rather than ex-post cash transfers, and improved early warnings was evaluated under different climate change scenarios"*
- L37 We added "potential" to better reflect the uncertain nature of this sentence
  *"Drought risk models are important tools to inform policy makers about the potential effectiveness of adaptation policies and …"*
- L180 We changed "robustness" for "effect" as this is a more applicable term in this context.
  *"While they not have a known probability of occurring, they enable testing the effect of the on-farm adaptations and top-down drought disaster risk reduction strategies on drought risk under changing average hydro-meteorological conditions."*
- L190 We added the explanation of the acronym (Generalized Extreme Value)
- L206 We changed multiple sentences (mainly the verbs) in order to better describe how interventions positively influence the intention to adapt of certain groups
  *"As shown in Wens et al (2021), extension services are most effective when offered to younger, less rich and less educated people, or to those who already adopted the most common measures. Similarly, early warning systems are changing the intention to adapt mostly for less educated, less rich farmers, or those not part of farmer knowledge exchange groups. The ex-ante cash transfer drives the adoption of more expensive measures for those who spend already a lot of money on adaptation, the most. Access to credit is preferred by less rich farmers, who have a larger land size, are members of a farm group, went to extension trainings, have easy access to information and/or are highly educated (Wens et al. 2021). "*
- L214 We clarified emergency aid is always given in the model, while the two "more than reactive" scenarios have additional interventions (emergency aid is not seen as a new intervention in the model – also the historic period has this). Since this wording is used throughout the text (reactive and no intervention) we decided to explain it rather than remove it.
  *"No (new, pro-active) interventions are implemented. Only emergency aid (standard in the ADOPT model to avoid households to die) is given to farmers who lost their livelihoods after drought disasters; this food aid is distributed to farmers who are on the verge of poverty to avoid famine."*
  Moreover, we explained the link between training and extension services.
  *"Besides, emergency services are provided in the form of frequent trainings given in communities with poor practices to improve their capacity related to drought adaptation practices for agriculture."*
- L255 We changed the y-axis as suggested to better visualize the differences of interventions.
- L286 We changed "with" to "to", as this is the better preposition to use in this context
- L287 We added "potential" to better reflect the uncertain nature of this sentence
  *"Clearly, an increased uptake of measures under this intervention scenario would potentially offset a potentially harmful drying climate trend."*
- L314 We elaborated the link between charcoal burning and poverty, adding a reference.
  *"It should be kept in mind that ADOPT does not consider (illicit) coping activities in the face of droughts which can – if a drought warning is send out – allow households to avoid buying food at high market prices or to engage in other income-generating activities such as food stocking or charcoal burning (Eriksen et al., 2005)."*

- L416 We added the explanation of the acronym (Agent-based models)
- L481 We removed "(non-)governmental" which was a relict from the previous revision round
- L501 We added a short sentence to sections 4.2, 4.3 and 5.1 to highlight the delayed effects

  L286: *"The adoption of adaptation measures by households influenced their maize yield and thus affected the average and median maize harvest under the different future climates and drought risk reduction interventions – with an increasing effect over the years (increasing difference in harvest between reactive and other scenarios, Fig. 6)."*

  L304: *"It is important to remark that the different between the intervention scenarios and the reactive scenario is only clearly visible after more than 10 years under most future climate scenarios."*

  L398: *"However, depending on the climate scenario applied, the effect of increased adoption due to a prospective interventions on household maize production, thus on food security and poverty, is only visible after a few years under drier conditions and after more than ten years under wetter conditions."*

Respectfully,

Marthe Wens